# ESYT1 tethers the ER to mitochondria and is required for mitochondrial lipid and calcium homeostasis

Alexandre Janer[1,2,*] , Jordan L Morris[3,*], Michiel Krols[4,5], Hana Antonicka[1,2] , Mari J Aaltonen[1,2] , Zhen-Yuan Lin[6,7], Hanish Anand[3], Anne-Claude Gingras[6,7] , Julien Prudent[3], Eric A Shoubridge[1,2]

Mitochondria interact with the ER at structurally and functionally specialized membrane contact sites known as mitochondria–ER contact sites (MERCs). Combining proximity labelling (BioID), co-immunoprecipitation, confocal microscopy and subcellular fractionation, we found that the ER resident SMP-domain protein ESYT1 was enriched at MERCs, where it forms a complex with the outer mitochondrial membrane protein SYNJ2BP. BioID analyses using ER-targeted, outer mitochondrial membrane-targeted, and MERC-targeted baits, confirmed the presence of this complex at MERCs and the specificity of the interaction. Deletion of ESYT1 or SYNJ2BP reduced the number and length of MERCs. Loss of the ESYT1–SYNJ2BP complex impaired ER to mitochondria calcium flux and provoked a significant alteration of the mitochondrial lipidome, most prominently a reduction of cardiolipins and phosphatidyl-ethanolamines. Both phenotypes were rescued by reexpression of WT ESYT1 and an artificial mitochondria–ER tether. Together, these results reveal a novel function for ESYT1 in mitochondrial and cellular homeostasis through its role in the regulation of MERCs.

## Introduction

Mitochondria interact with several membrane-delimited organelles within the cell, including the ER, lysosomes, peroxisomes, and trans-Golgi network vesicles (Tabara et al, 2021). Mitochondria–ER contact sites (MERCs), also called mitochondria-associated membranes (MAMs) when studied at a biochemical level, are the best characterized class of membrane contact sites (MCSs) and represent the close apposition of the outer mitochondrial membrane (OMM) with the ER membrane (Giacomello & Pellegrini, 2016). MERCs are functionally and structurally specialized cellular sub-domains that form signaling platforms allowing lipid synthesis and

transport, calcium signalling, apoptosis regulation, mitochondrial division, and autophagosome formation (Herrera-Cruz & Simmen, 2017; Giacomello et al, 2020). MERCs have also been shown to be involved in several critical cellular pathways such as metabolic regulation in diabetes (Rieusset, 2017), inflammation (Missiroli et al, 2018), the immune response (Martinvalet, 2018), and senescence (Janikiewicz et al, 2018). Alterations in these structures have also been linked to the onset of neurodegenerative diseases including Alzheimer's disease, Parkinson's disease, and amyotrophic lateral sclerosis (Vallese et al, 2020), and aging (Janikiewicz et al, 2018).

The proteins that mediate the formation of MERC have been extensively studied in the yeast Saccharomyces cerevisiae, where the four-subunit ER–mitochondria encounter structure is required to tether the two organelles and mediate lipid transport from the ER to mitochondria (Kornmann et al, 2009; Kojima et al, 2016) via the lipid-binding SMP domains (synaptotagmin-like mitochondrial and lipid-binding protein) present in three subunits of the complex (Kopec et al, 2010; AhYoung et al, 2015). Orthologues of the three SMP domain-containing proteins in the ER–mitochondria encounter structure complex have not been identified in mammals.

Mitochondria synthesize cardiolipin (CL) and phosphatidyleth-anolamine (PE) on the inner membrane, and these lipids are essential for mitochondrial function (Steenbergen et al, 2005; Funai et al, 2020). CL is produced via a multi-enzymatic cascade and PE is synthesized by phosphatidylserine decarboxylase PISD1; however, their synthesis depends on the ER for the supply of the precursor lipids phosphatidic acid (PA) and phosphatidylserine (PS), respectively (Funai et al, 2020). Lipid synthesis activity at MAMs was the first biochemical process reported at a MCS in mammals (Vance, 1990); however, a detailed mechanism of lipid transport between ER and mitochondria in mammals remains elusive.

All SMP domain-containing proteins are present at MCSs, where they are thought to facilitate non-vesicular transport of lipids between lipid bilayers (Jeyasimman & Saheki, 2020). In mammals, the ER-anchored extended synaptotagmin (ESYT) proteins are the

[1]Department of Human Genetics, McGill University, Montreal, Canada   [2]Montreal Neurological Institute, McGill University, Montreal, Canada   [3]Medical Research Council Mitochondrial Biology Unit, University of Cambridge, Cambridge, UK   [4]Montreal Neurological Institute, McGill University, Montreal, Canada   [5]Department of Neurology and Neurosurgery, McGill University, Montreal, Canada   [6]Lunenfeld-Tanenbaum Research Institute, Mount Sinai Hospital, Toronto, Canada   [7]Department of Molecular Genetics, University of Toronto, Toronto, Canada

Correspondence: eric.shoubridge@mcgill.ca
Hanish Anand's present address is Hepatocyte Biology and Transplantation Group, Institute of Liver Studies, King's College Hospital, London, UK
*Alexandre Janer and Jordan L Morris contributed equally to this work

best characterized (Saheki & De Camilli, 2017). ESYT1, ESYT2, and ESYT3 tether the ER to the plasma membrane (PM), potentially transferring lipids (Bian et al, 2018). More specifically, ESYT1 has been shown to play a role in $Ca^{2+}$-dependent lipid transfer at ER–PM contacts, which requires its docking with $PIP(4,5)P_2$ in the plasma membrane (Giordano et al, 2013; Reinisch & De Camilli, 2016; Bian et al, 2018; Ge et al, 2022). It also tethers ER to peroxisomes by a similar mechanism facilitating the transport of cholesterol (Xiao et al, 2019), raising the possibility that ESYT1 could also tether ER to mitochondria to promote lipid transfer.

In this study, we used the proximity mapping tool BioID to identify and characterize SMP domain proteins that might be involved in MERC structure and function in humans. We showed that ESYT1 is enriched at MERCs, where it forms a complex with the OMM protein SYNJ2BP. Depletion of the ESYT1–SYNJ2BP complex impairs mitochondrial calcium uptake capacity and provokes a reduction of essential mitochondrial lipids, demonstrating its essential function in cellular and mitochondrial homeostasis.

# Results

### Proximity labelling analysis of SMP domain proteins in human cells

We recently established that the proximity of proteins localized on two different membrane-bound organelles can be detected by the proximity mapping tool BioID (Antonicka et al, 2020; Go et al, 2021). To identify potential proteins involved in the regulation of MCSs and lipid transport between ER and mitochondria, we selected several ER-resident human SMP domain-containing proteins as baits (PDZD8, TEX2, ESYT2, and ESYT1). We generated stable inducible Flp-In T-REx 293 cell lines expressing each protein fused with BirA* (Fig S1A) and used BioID to characterize their proximity interactomes and identify potential interacting partners on the OMM.

BioID analysis of the selected SMP domain-containing proteins (Table S1) revealed that, as expected, most of their proximity interactors were ER membrane proteins involved in organelle organization, transport, lipid biosynthesis, and metabolic regulation. (34 of 40 preys shared among all four baits were ER proteins, Fig S1B and Table S1). Each bait also detected numerous unique proximity interactors. In addition, two preys common to all four baits, ALDH3A2 and FKBP8, have been reported to dually localize to mitochondria and ER (Shirane & Nakayama, 2003; Rath et al, 2021; Zeng et al, 2021) (Fig S1B).

PDZD8 was previously shown to partially localize to MERCs and tether the two organelles (Hirabayashi et al, 2017), but its interacting partner on the OMM remains unknown. Because of its capacity to regulate MERCs, the absence of PDZD8 led to decreased mitochondrial calcium uptake capacity upon ER stimulation (Hirabayashi et al, 2017). PDZD8 was later described to interact with RAB7 and ZFYVE27 (Protrudin) to establish three-way MCSs between the ER, late endosomes, and mitochondria and to mediate lipid transfer required for late endosome maturation (Elbaz-Alon et al, 2020; Shirane et al, 2020; Khan et al, 2021; Gao et al, 2022). Mass spectrometry results

obtained with either the N- or C-terminal PDZD8-BirA* fusion proteins confirmed the proximity interaction with ZFYVE27 but failed to identify any OMM-localized partner (Table S1).

TEX2 is still uncharacterized in mammals; however, its yeast ortholog Nvj2 localizes to ER–vacuole (lysosome-like organelle) contact sites at steady state. Upon ER stress or ceramide overproduction, it translocates to ER–Golgi contacts to facilitate the non-vesicular transport of ceramide from the ER to the Golgi, counteracting ceramide toxicity (Liu et al, 2017). Consistent with the role of Nvj2 in yeast, we identified 12 proteins belonging to the ER–Golgi vesicle-mediated transport pathway in the TEX2 proximity interactome (Table S1, in green); however, as with PDZD8, we did not identify an OMM proximity interactor.

In contrast to ESYT2, that constitutively tethers ER to the PM and is localized in the cortical ER, the interaction of ESYT1 with the PM is activated by $Ca^{2+}$ binding. The proportion of ESYT1 present throughout the ER or concentrated at ER–PM contacts is controlled by cytosolic $Ca^{2+}$ (Chang et al, 2013; Giordano et al, 2013; Idevall-Hagren et al, 2015). As ESYT members could form heteromeric complexes, ESYT-dependent ER–PM contacts are regulated by both cytosolic $Ca^{2+}$ and the specific phospholipid $PI(4,5)P_2$ at the PM (Fernandez-Busnadiego et al, 2015). In both N- and C-terminal ESYT1-BirA* experiments (Table S1), we confirmed the interaction with its known partner ESYT2. Importantly, we also found a unique specific proximity interaction with the OMM protein SYNJ2BP (OMP25) (Figs 1A and S1B). This interaction was previously noted but never further investigated (Christianson et al, 2011; Hung et al, 2017). Significantly, ESYT2 BioID analysis also identified ESYT1 (Table S1) as its main proximity interactor but failed to identify SYNJ2BP, suggesting that ESYT1 may form a specific complex with SYNJ2BP at MERCs independent of its interaction with ESYT2 at ER–PM contacts.

Immunoprecipitation of ESYT1 from human fibroblasts stably overexpressing a C-terminal 3xFLAG-tagged version of ESYT1 followed by LC–MS analysis showed that SYNJ2BP (and ESYT2) co-immunoprecipitated with ESYT1 (Table S2), confirming our proximity interaction results.

We further compared the BioID profile of SMP proteins with the BioID of an ER-targeted BirA*, that promiscuously labels proteins in the ER and vicinity, serving as a control for protein-independent ER proximity labelling (Table S1). SYNJ2BP was not found as proximity interactor of ER-BirA*, further validating the specificity of the interaction between ESYT1 and SYNJ2BP (Fig S1C and D).

These data prompted us to perform a BioID analysis using SYNJ2BP as bait (Fig S1A and Table S1) and we observed a strong enrichment of ESYT1, confirming the proximity interaction of the two partners. SYNJ2BP was shown to interact with another ER-localized protein RRBP1 to regulate the formation of MERCs (Hung et al, 2017), and we also identified RRBP1 as prey. Hung et al (2017) also reported an interaction between SYNJ2BP and the multi aminoacyl tRNA synthetase complex (Mirande, 2017), an interaction we also confirmed, further substantiating the specificity of our BioID results.

We then compared the BioID profile of SYNJ2BP with the BioID of an OMM-targeted BirA*, serving as a control for protein-independent OMM proximity labelling (Table S1). ESYT1 was not found as proximity interactor of OMM-BirA*, validating the specificity of the interaction between ESYT1 and SYNJ2BP (Table S1 and Fig 1B).

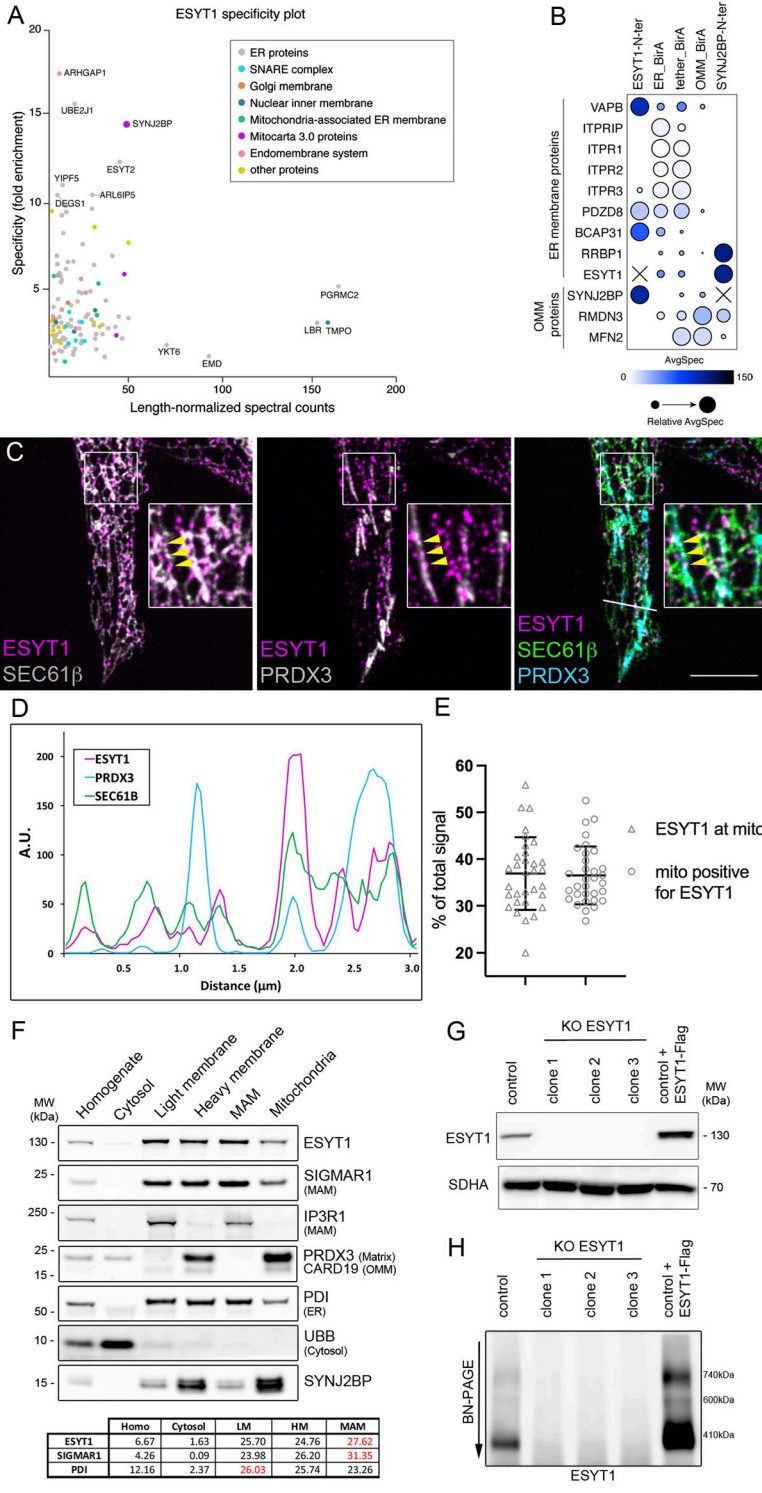

**Figure 1. ESYT1 localizes to mitochondria–ER contact sites where it interacts with SYNJ2BP.**
**(A)** Specificity plot of ESYT1-N-ter BioID analysis indicates the specific proximity interaction with SYNJ2BP. The specificity denotes the fold enrichment of the spectral counts detected for each prey in the ESYT1 BioID compared with the spectral counts for that prey in all other baits in the dataset (all four SMP proteins). Prey names for the most specific preys and for preys with the highest length-normalized spectral counts are indicated. Preys are colour-coded based on their GO term cellular compartment analysis. MitoCarta3.0 proteins are SYNJ2BP, FKBP8, and ALDH3A2. **(B)** Proximity interaction between known (and predicted) ER–mitochondrial tethers with indicated baits (BFDR ≤ 0.01). The colour of each circle represents the prey-length normalized average spectra detected for the indicated protein by each bait and the size of the circle represents the relative average spectra across the baits analyzed in this dataset. The SAINT analysis excludes self-detection for the bait protein as a prey, and is represented as X in the graph. **(C)** Confocal microscopy images of endogenous ESYT1 localization (magenta) in human fibroblasts stably overexpressing SEC61B-mCherry as an ER marker (green). Staining for endogenous PRDX3 serves as a mitochondrial marker (cyan). Yellow arrows point to foci of ESYT1 colocalizing with both ER and mitochondria. Scale bar = 5 μm. **(D)** Line scan of fluorescence intensities demonstrating focal accumulations of endogenous ESYT1 along the ER network that partially colocalize with mitochondria (A.U. = arbitrary units). **(E)** Quantitative confocal microscopy analysis of endogenous ESYT1 localization in control human fibroblasts stably overexpressing SEC61B-mCherry as an ER marker, labelled with ESYT1 and with TOMM40 as a mitochondrial marker. Percentage of ESYT1 signal colocalizing with mitochondria and percentage of mitochondria positive for ESYT1 were assessed. Results are expressed as means ± S.D. (n = 32). **(F)** Subcellular localization of endogenous ESYT1 and SYNJ2BP. Mouse liver was fractionated, and the fractions were analyzed by SDS–PAGE and immunoblotting. SIGMAR1 and IP3R1 are MAM markers, PRDX3 is a mitochondrial matrix marker, CARD19 is an outer mitochondrial membrane marker, PDI is an ER marker, and UBB is a cytosol marker. The percentage of ESYT1, SIGMAR1, and PDI signal in each fraction is shown. **(G)** ESYT1 protein levels in control human fibroblast, three individual clones of ESYT1 knock-out fibroblasts and fibroblasts overexpressing ESYT1-3xFLAG. Whole-cell lysates were analyzed by SDS–PAGE and immunoblotting. SDHA was used as a loading control. **(H)** Characterization of the ESYT1 complexes. Heavy membrane fractions were isolated from control human fibroblasts, ESYT1 knock-out fibroblasts, and fibroblasts overexpressing ESYT1-3xFLAG, solubilized with 1% DDM and analyzed by blue native PAGE.

In conclusion, of the four SMP domain-containing proteins we profiled, only ESYT1 identified a specific proximity interacting partner on the OMM, SYNJ2BP, suggesting that this complex could play a role in the regulation of MERC formation and/or function.

## ESYT1 localizes to MERCs

To further investigate the interaction between ESYT1 and SYNJ2BP at MERCs, we profiled the proximity interactome of the MERCs using an

engineered MERC-targeted BirA* (Fig S1E and Table S1). This construct was based on a fluorescent MERC tether first designed by Hajnoczky (Csordas et al, 2006) and reported to successfully rescue both MERC and $Ca^{2+}$ loss in cells devoid of several other contact site protein regulators including inositol-3-phosphate receptor (IP3R), PDZD8, RMDN3-VAPB or MFN2 (Gomez-Suaga et al, 2017; Hirabayashi et al, 2017; Hernández-Alvarez et al, 2019). BirA* was then fused between the OMM-targeting sequence of mAKAP1 at the N-terminus and the ER-targeting sequence of yUBC6 at the C-terminus. We analysed the tether-BirA* proximity interactions with previously characterized MERC proteins alongside ESYT1 and SYNJ2BP (Fig 1B) and showed that tether-BirA* interacted with all the queried preys, consistent with an interaction of ESYT1 and SYNJ2BP at MERCs.

To confirm this localization, we next studied ESYT1 intracellular localization by immunofluorescence and confocal microscopy (Fig 1C). In human fibroblasts stably overexpressing SEC61B-mCherry as an ER marker (green) and stained for PRDX3 as a mitochondrial marker (cyan), endogenous ESYT1 (magenta) specifically localized along the ER network forming puncta, especially on ER tubules (which function in lipid and hormone synthesis) rather than on the perinuclear sheets (which function in protein synthesis) (Schwarz & Blower, 2016). The focal localization of endogenous ESYT1 along the ER network partially colocalized with mitochondria (Fig 1C, yellow arrows), illustrated by line scans of fluorescence intensities (Fig 1D). Quantitative analysis confirmed that more than 30% of the endogenous ESYT1 colocalized with mitochondria and that a third of mitochondria were positive for ESYT1 (Fig 1E).

Consistent with these results, subcellular fractionation of mouse liver (Fig 1F) showed that endogenous ESYT1 is present in the microsomal light membrane fraction containing ER, and in the heavy membrane fraction containing mitochondria and MAM. Gradient-purification of the heavy membranes into MAM and highly purified mitochondria revealed that ESYT1 was enriched in MAMs, with a similar fractionation profile as the MAM marker SIGMAR1. Significantly, SYNJ2BP, in addition to being enriched in mitochondria, was also present in the MAM fraction.

To further characterize the function of ESYT1, we generated a CRISPR-Cas9–mediated KO in human fibroblasts and fibroblasts stably overexpressing a C-terminal 3xFLAG-tagged version of ESYT1 (Fig 1G). BN-PAGE analysis of DDM-solubilized heavy membrane fractions (Fig 1H) revealed that endogenous ESYT1 was present in three main large complexes, with the main one at approximately 410 kD. The specificity of these complexes was confirmed by their absence in different clones of the KO cell lines. Finally, the ESYT1-FLAG overexpressing cell line showed that the tagged version of ESYT1 behaved similarly to the endogenous protein (Fig 1H), but formed slightly larger complexes because of the addition of the 3xFLAG tag.

Together, these results show that ESYT1 and its OMM partner SYNJ2BP localize to the MERCs, and that ESYT1 forms high molecular weight complexes.

## Loss of ESYT1 decreases MERCs

As ESYT1 is known to tether the ER membrane to the PM (Saheki, 2017) and to peroxisomes (Xiao et al, 2019), we sought to determine whether ESYT1 could similarly act as a tethering protein regulating MERCs. Using transmission electron microscopy (TEM), we analyzed the morphology and characteristics of MERCs in human control fibroblasts compared with ESYT1 KO cells and KO cells where a Myc-tagged version of ESYT1 was stably reintroduced (Fig 2A). TEM image analysis revealed that the loss of ESYT1 led to a decrease in both the number and mean length of MERCs, resulting in an overall decrease in the perimeter of mitochondria covered by ER membrane (Fig 2B and C). MERC defects were completely rescued by the reintroduction of ESYT1–Myc, confirming the specificity of this phenotype. Notably, mitochondria in ESYT1 KO cells have a larger perimeter than control cells, a phenotype that was fully rescued by the expression of ESYT1–Myc. The larger perimeter likely results from the loss of MERCs, which demarcate sites of mitochondrial fission (Giacomello et al, 2020). These experiments show that loss of ESYT1 impacts MERC formation, and suggests a potential direct role as a physical tether between the two organelles.

## SYNJ2BP but not ESYT1 promotes the formation of mitochondria–ER contacts

We next investigated the consequences of the overexpression of ESYT1, or its mitochondrial partner SYNJ2BP on MERC architecture. The overexpression of a 3xFLAG-tagged version of ESYT1 did not influence the morphology of MERCs (Fig 3A and B); however, as was previously demonstrated (Nemoto & De Camilli, 1999; Hung et al, 2017; Pourshafie et al, 2022), SYNJ2BP overexpression strikingly promoted the formation of MERCs, specifically by increasing the length of individual contacts between the two organelles and the mitochondrial perimeter in contact with the ER in a "zipper-like" fashion (Fig 3B). In this condition, the perimeter of mitochondria was smaller and the ER–mitochondrial network was recruited to the perinuclear region of the SYNJ2BP overexpressing cells (Fig 3A). Immunofluorescence and confocal microscopy analysis confirmed both the significant increase of MERCs and the perinuclear accumulation of the ER–mitochondrial network when SYNJ2BP was overexpressed (Fig S2A). In these conditions, we also observed that endogenous ESYT1 was recruited to MERCs, where it accumulated and formed large *foci* (Fig S2A, white arrowheads). Quantitative analysis, using confocal microscopy to compare control, SYNJ2BP KO, and SYNJ2BP overexpressing fibroblasts, demonstrated that the presence of ESYT1 at mitochondria is dependent on SYNJ2BP expression (Fig 3C). In contrast to SYNJ2BP overexpression, loss of SYNJ2BP which decreased MERCs (Ilacqua et al, 2022; Pourshafie et al, 2022) was associated with a decreased localization of ESYT1 at mitochondria.

SYNJ2BP was shown to interact with another ER-localized protein RRBP1 to regulate the formation of MERCs (Hung et al, 2017). To explore the relation between SYNJ2BP, ESYT1 and RRBP1, we analyzed their subcellular localization in human control fibroblasts and fibroblasts overexpressing SYNJ2BP (Fig 3D). Although ESYT1 and RRBP1 are both ER membrane proteins, their localization differs in control cells. RRBP1 is preferentially localized on the perinuclear sheets and ESYT1 on ER tubules (Fig 3D(a)). When SYNJ2BP is overexpressed and MERCs increased, the large ESYT1 foci recruited to mitochondria specifically localize in regions of SYNJ2BP accumulation (Fig 3D(b) white arrowheads). In addition, we observed a mitochondrial ghost pattern for ESYT1 localization that we do not

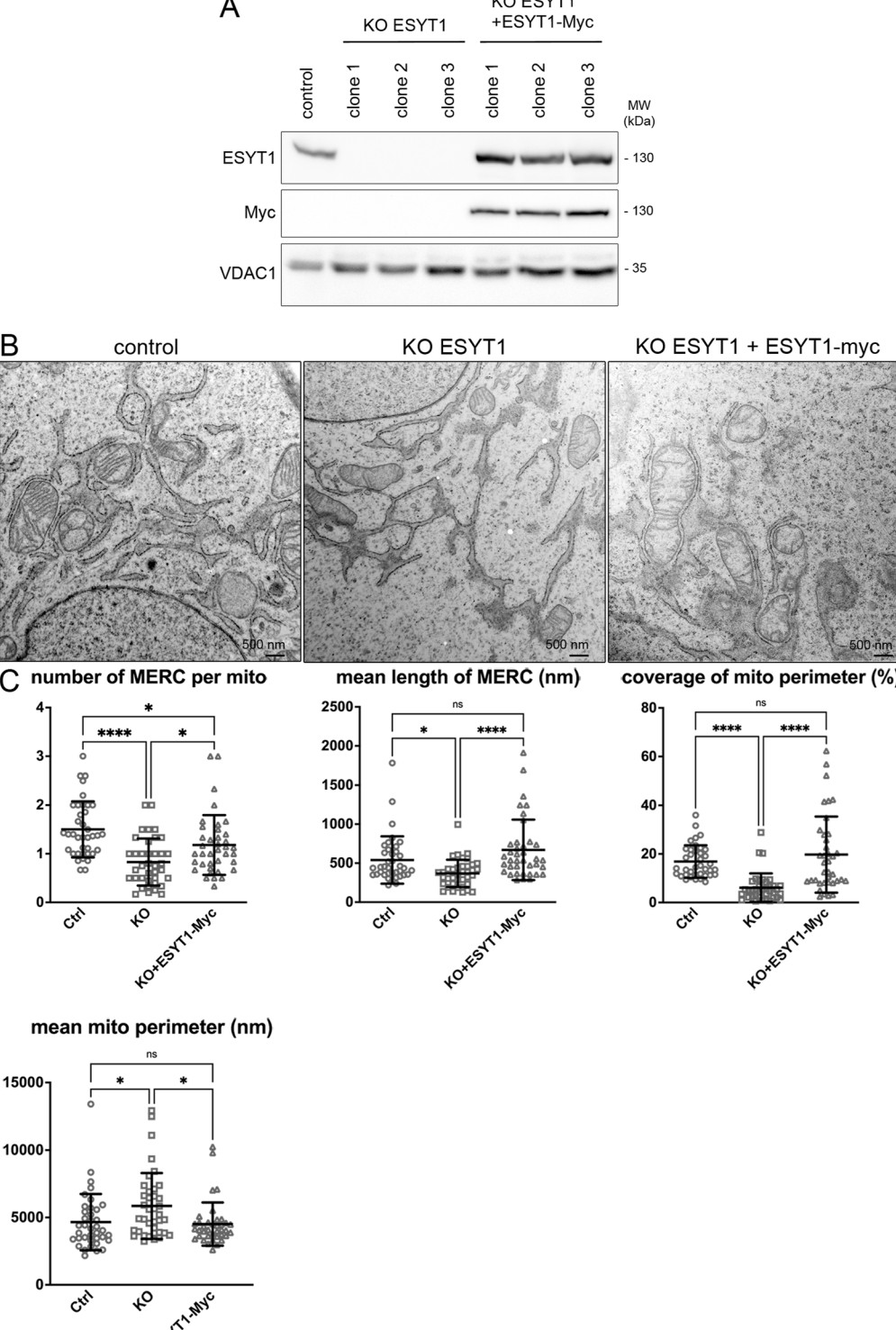

**Figure 2. Loss of ESYT1 decreases MERCs.**
**(A)** ESYT1 protein levels in control human fibroblasts, ESYT1 knock-out fibroblasts, and ESYT1 knock-out fibroblasts expressing ESYT1-Myc. Whole cell lysates were analyzed by SDS–PAGE and immunoblotting. VDAC1 was used as a loading control. **(B)** Transmission electron microscopy images of control human fibroblasts, ESYT1 knock-out fibroblasts, and ESYT1 knock-out fibroblasts expressing ESYT1-Myc. **(C)** Quantitative analysis of Mitochondria–ER contact sites (MERCs) from the TEM images: number of MERC per mitochondria, length of MERC (nm), coverage of the mitochondrial perimeter by ER (%), and mitochondrial perimeter (nm). Results are expressed as means ± S.D. Images in each condition were analyzed (n = 38), totaling 245 mitochondria for control cells, 154 mitochondria for KO cells, and 224 mitochondria for rescued cells. Kruskal–Wallis and post hoc multiple comparisons tests were applied, ns: nonsignificant, *P < 0.05, ****P < 0.0005.

see in control cells (Fig 3D(c) yellow arrowheads). In this condition, ESYT1 and RRBP1 actually accumulate in different areas of the mitochondrial network (Fig 3D(c)), suggesting different functions of the SYNJ2BP–ESYT1 and SYNJ2BP–RRBP1 complexes. Quantitative

confocal microscopy analysis of MERCs in control fibroblasts, SYNJ2BP overexpressing fibroblasts, ESYT1 KO fibroblasts, and ESYT1 KO fibroblasts overexpressing SYNJ2BP (Fig 3E) confirmed the reduction of MERCs in the absence of ESYT1 and showed that the

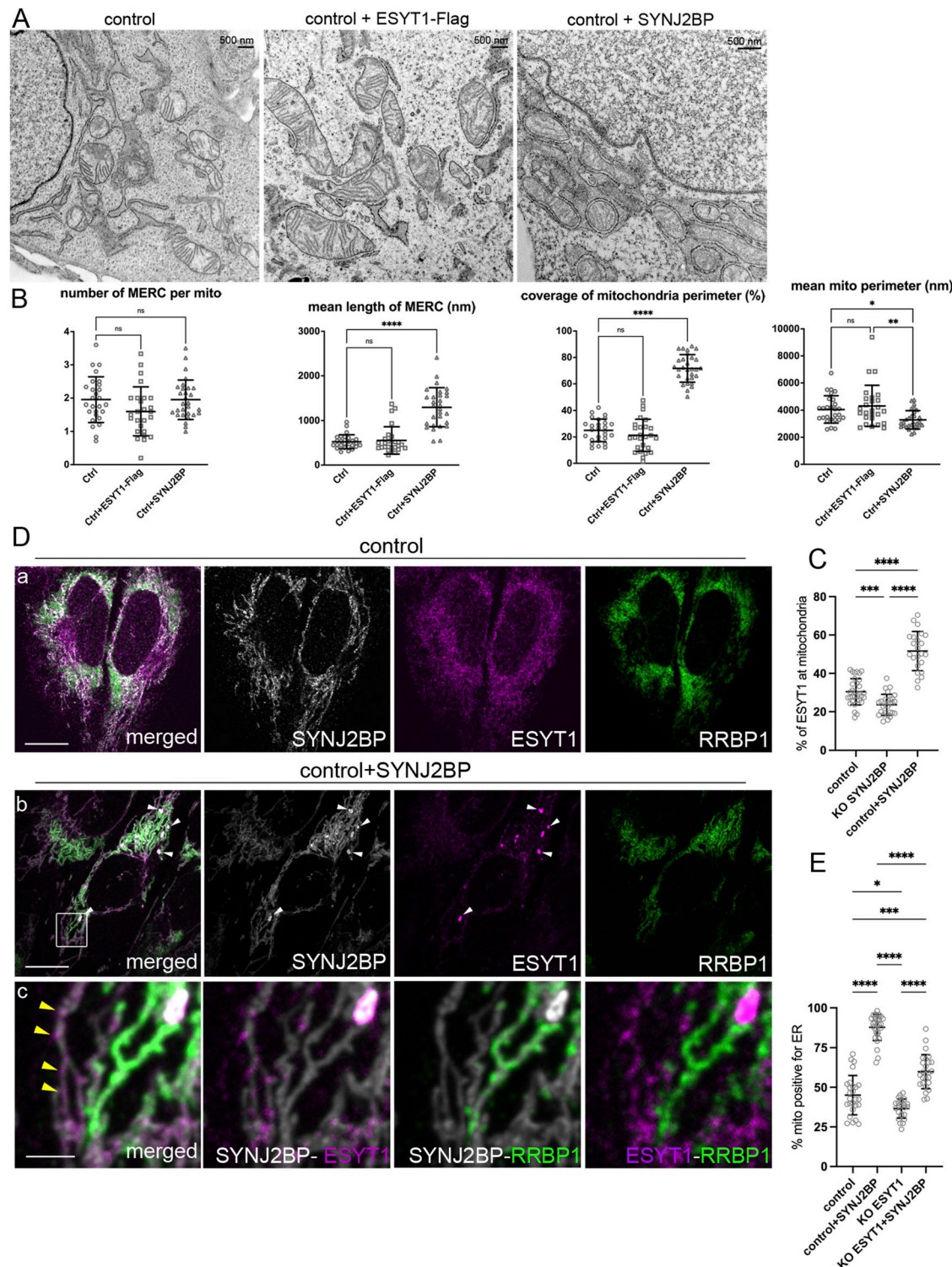

**Figure 3. SYNJ2BP but not ESYT1 promotes the formation of mitochondria–ER contacts.**

**(A)** Transmission electron microscopy images of control human fibroblasts, fibroblasts overexpressing ESYT1-FLAG, and fibroblasts overexpressing SYNJ2BP.
**(B)** Quantitative analysis of mitochondria–ER contact sites (MERCs) in control human fibroblasts, fibroblasts overexpressing ESYT1-FLAG, and fibroblasts overexpressing SYNJ2BP showing the number of MERC per mitochondria, the length of MERC (nm), and the coverage of the mitochondrial perimeter by ER (%), and mitochondrial perimeter (nm). Results are expressed as means ± S.D. Images were analyzed in control fibroblasts (n = 27), totaling 152 mitochondria; in fibroblasts overexpressing ESYT1-FLAG (n = 26), totaling 140 mitochondria; in fibroblasts overexpressing SYNJ2BP (n = 29), totaling 300 mitochondria. Kruskal–Wallis and post hoc multiple comparisons tests were applied, ns: nonsignificant, *P < 0.05, **P < 0.01, ****P < 0.0005. **(C)** Quantitative confocal microscopy analysis of endogenous ESYT1 colocalization with mitochondria in control human fibroblasts (n = 32), SYNJ2BP KO fibroblasts (n = 28), and fibroblasts overexpressing SYNJ2BP (n = 23). Cells were labelled with ESYT1 and PRDX3 as a

effect of SYNJ2BP overexpression on MERC formation is partially dependant on the presence of ESYT1.

Because of the contribution of mitochondrial fission and fusion-related proteins in the formation or stabilization of MERCs, including the OMM fusion protein MFN2 (de Brito & Scorrano, 2008) and the main mitochondrial fission regulator DRP1 (Prudent et al, 2015), we decided to investigate their potential contribution to SYNJ2BP-dependent MERC formation. Control cells and cells overexpressing SYNJ2BP were depleted for either DRP1 or MFN2 (Fig S2B). As expected, in both control cells and cells overexpressing SYNJ2BP, depletion of DRP1 led to a hyperfused mitochondrial network (a and b), whereas loss of MFN2 induced mitochondrial fragmentation (c and d). In both conditions, the overexpression of SYNJ2BP still promoted a strong increase of MERCs as monitored by confocal microscopy (b and d, cyt *c* as a mitochondrial marker and HSPA5 as an ER marker). However, the recruitment of the ER–mitochondrial network around the nucleus was less prominent after DRP1 knockdown. We conclude that the effect of SYNJ2BP on MERC formation is independent of MFN2 and DRP1.

## SYNJ2BP is present in a high-molecular weight complex with ESYT1

To better understand the relationship between ESYT1 and SYNJ2BP, we investigated their potential interaction by BN-PAGE analysis. Whereas endogenous SYNJ2BP ran mostly as a monomer (Fig 4A, left), when overexpressed (a condition that promotes MERCs), SYNJ2BP appeared in two high molecular weight complexes (Fig 4A, left), one of which was at the same size as the ESYT1 complex at 410 kD (Fig 4A, right, lower horizontal line). Overexpression of SYNJ2BP together with a 3xFLAG tagged version of ESYT1 leads to the shift of ESYT1 complex to a higher molecular weight. In this condition, the 410 kD SYNJ2BP complex specifically shifted to a similar molecular weight, demonstrating the interaction of the two partners in this complex (Fig 4A, right, higher horizontal line). A second dimension BN/SDS–PAGE analysis confirmed that when overexpressed, a fraction of SYNJ2BP is present in two different complexes, one that runs at the size of the ESYT1 complex and one to similar size of the RRBP1 complex (Fig 4B). Knockdown of RRBP1 did not affect the assembly of ESYT1 complex (Fig 4C), nor did the knockdown of ESYT1 affect the RRBP1 complex, demonstrating that the complexes are not interdependent. However, the presence of SYNJ2BP in the 410 kD complex is specifically dependant on ESYT1, because its depletion leads to the loss of the SYNJ2BP complex at 410 kD (Fig 4C), demonstrating that ESYT1 and SYNJ2BP belong to the same complex.

A study by Hung et al reported that the interaction of SYNJ2BP with RRBP1 depends on cytoplasmic translation activity (Hung et al, 2017). To confirm that the two SYNJ2BP complexes are independent, we analyzed the effects of puromycin, a translation inhibitor, on the formation of both complexes. Puromycin treatment led to a large decrease in the steady-state level of RRBP1 and a concomitant increase of ESYT1, without affecting SYNJ2BP levels (Fig 4D). A second-dimension experiment confirmed that puromycin induced a specific loss of the SYNJ2BP–RRBP1 complex, without affecting the complex between SYNJ2BP and ESYT1 (Fig 4E). Together, these results demonstrate that SYNJ2BP interacts with both ESYT1 and RRBP1, but in two different complexes that are physically and functionally independent.

## ESYT1 is required for ER to mitochondria Ca²⁺ transfer

In mammals, the best characterized functional feature of MERCs is $Ca^{2+}$ flux from the ER to mitochondria required to sustain mitochondrial homeostasis (Rossi et al, 2019). $Ca^{2+}$ is released from the ER through the IP3R and crosses the OMM through the voltage-dependent anion channel, which interacts with IP3R via the cytosolic protein GRP75 (Szabadkai et al, 2006). $Ca^{2+}$ is then transported to the matrix via the IMM mitochondrial calcium uniporter (MCU) complex (De Stefani, Raffaello et al, 2011; Bick et al, 2012). MERCs provide spatially constrained microdomains in which $Ca^{2+}$ released from the ER can accumulate at high concentrations sufficient to induce mitochondrial $Ca^{2+}$ uptake via the low $Ca^{2+}$ affinity MCU (Rizzuto et al, 1998; Csordas et al, 2006; Szabadkai et al, 2006). As a consequence, proteins that regulate MERC formation affect ER to mitochondria $Ca^{2+}$ transfer; a decrease of MERCs has been widely associated to a decrease of $Ca^{2+}$ transfer from the ER to mitochondria (de Brito & Scorrano, 2008; De Vos, Morotz et al, 2012; Stoica et al, 2014; Hirabayashi et al, 2017).

ER–PM contact sites are responsible for store-operated $Ca^{2+}$ entry (SOCE), a process allowing cellular, and in particular, cytosolic and ER, $Ca^{2+}$ replenishment (Ahmad et al, 2022). Silencing ESYT1 impairs SOCE efficiency in Jurkat cells (Woo et al, 2020), but not in HeLa cells (Giordano et al, 2013; Woo et al, 2020). To avoid confounding effects because of the loss of ESYT1 at ER–PM, and to SOCE impairment which can impact mitochondrial $Ca^{2+}$ uptake capacity, we first evaluated mitochondrial $Ca^{2+}$ pumping upon ER-$Ca^{2+}$ release in HeLa cells (Fig 5). We compared control cells, ESYT1 knock-down cells, and ESYT1 knock-down cells expressing an engineered ER–mitochondria tether (Hirabayashi et al, 2017). Knock-down of ESYT1 led to a decrease of mitochondrial $Ca^{2+}$ uptake from the ER upon histamine stimulation, as monitored by a genetically encoded $Ca^{2+}$ indicator targeted to the mitochondrial matrix (CEPIA-2mt) (Suzuki et al, 2014) (Fig 5A and B). Importantly, the expression of the artificial mitochondria–ER tether was able to rescue mitochondrial $Ca^{2+}$ defects observed in ESYT1 silenced cells upon histamine

---

mitochondrial marker. Results are expressed as means ± S.D. \*\*\*$P < 0.0005$; \*\*\*\*$P < 0.0001$ (Brown–Forsythe and Welch ANOVA test). **(D)** Confocal microscopy images of control human fibroblasts (a) and fibroblasts overexpressing SYNJ2BP (b, c) showing SYNJ2BP localization (grey), ESYT1 localization (magenta), and RRBP1 localization (green). White arrows point to large foci of endogenous ESYT1 colocalizing with SYNJ2BP accumulations when SYNJ2BP is overexpressed. Scale bar = 10 $\mu$m. (c): zoomed image from (b) showing ESYT1 and RRBP1 accumulation in different mitochondria when SYNJ2BP is overexpressed. Yellow arrowheads point to mitochondrial ghost pattern for ESYT1 localization when SYNJ2BP is overexpressed. Scale bar = 2 $\mu$m. **(E)** Quantitative confocal microscopy analysis of mitochondria positive for ER in control human fibroblasts (n = 26), SYNJ2BP overexpressing fibroblasts (n = 29), ESYT1 KO fibroblasts (n = 24) and ESYT1 KO fibroblasts overexpressing SYNJ2BP (n = 26). Cells were labelled with PRDX3 as a mitochondrial marker and CANX as an ER marker. Results are expressed as means ± S.D. \*$P < 0.05$; \*\*\*$P < 0.0005$; \*\*\*\*$P < 0.0001$ (Brown–Forsythe and Welch ANOVA test).

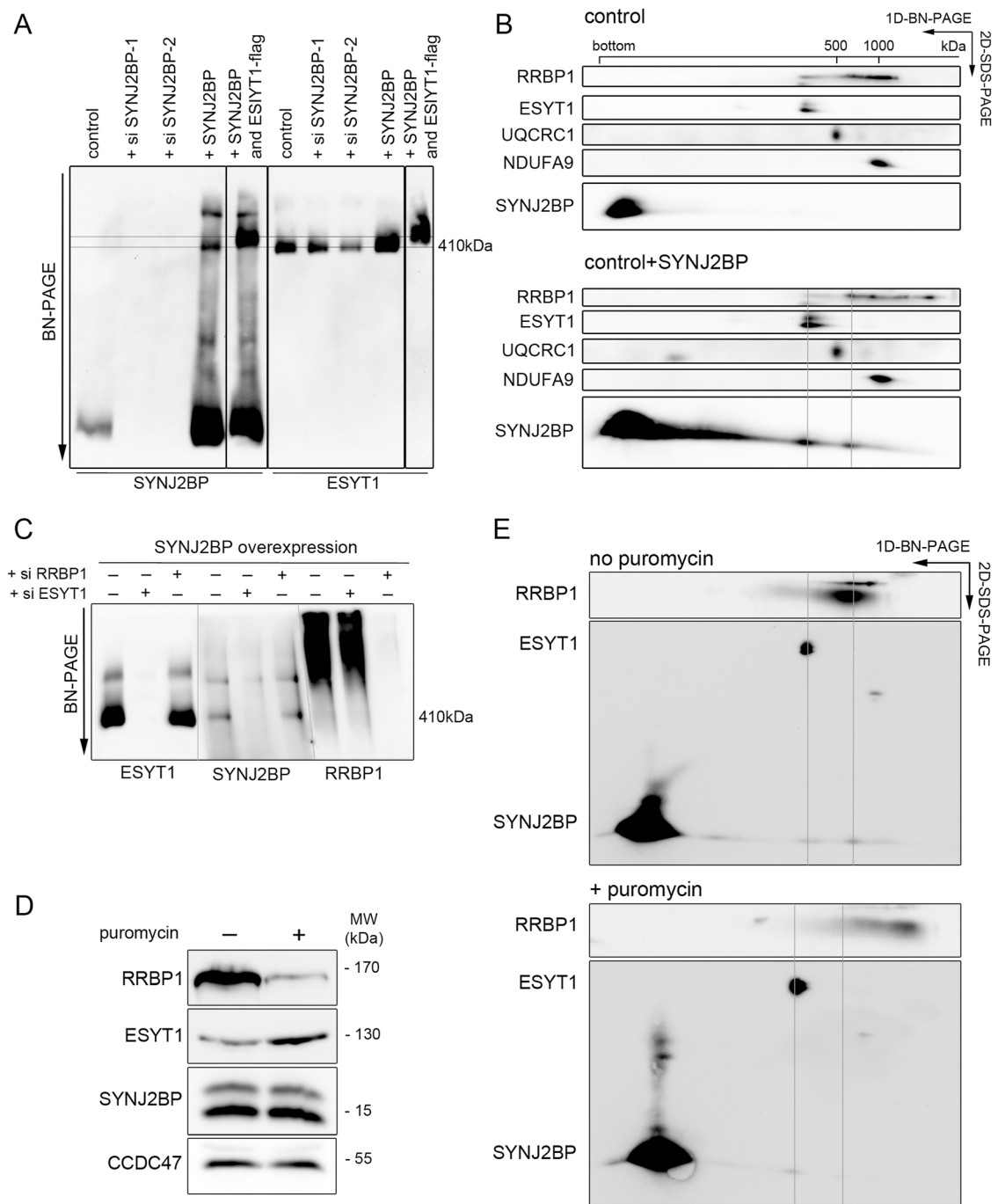

**Figure 4.  SYNJ2BP is present in a high-molecular weight complex with ESYT1.**
**(A)** Characterization of ESYT1 and SYNJ2BP complexes. Heavy-membrane fractions from control human fibroblasts, SYNJ2BP knock-down fibroblasts, fibroblasts overexpressing SYNJ2BP, and fibroblasts overexpressing SYNJ2BP together with a 3XFLAG-tagged version of ESYT1 were analyzed by blue native PAGE. Samples were run in duplicate on the same gel and immunoblotted with anti-SYNJ2BP (left) and anti-ESYT1 antibodies (right). Lower horizontal line: 410 kD complex where both SYNJ2BP and ESYT1 run. Higher horizontal line: higher molecular weight complex observed when SYNJ2BP is overexpressed together with a 3xFLAG-tagged version of ESYT1. **(B)** Two-dimensional electrophoresis analysis (BN-PAGE/SDS–PAGE) of SYNJ2BP-interacting proteins in control human fibroblasts and fibroblasts overexpressing SYNJ2BP. The migration of known protein complexes in the first dimension is indicated on the top of the blot (UQCRC1: OXPHOS complex III at 500 kD, NDUFA9: OXPHOS complex I at 1,000 kD). The position of identified SYNJ2BP containing complexes and their alignment with ESYT1 and RRBP1 containing complexes are indicated with grey lines. **(C)** Characterization of ESYT1, SYNJ2BP, and RRBP1 complexes. Heavy-membrane fractions from fibroblasts overexpressing SYNJ2BP or fibroblasts overexpressing SYNJ2BP in which either ESYT1 or RRBP1 was knocked down were analyzed by Blue-Native PAGE. Samples were run in triplicate on the same gel and immunoblotted with anti-ESYT1 (left), anti-SYNJ2BP (center), and anti-RRBP1 antibodies (right). **(D)** RRBP1, ESYT1 and SYNJ2BP protein levels in fibroblasts overexpressing SYNJ2BP untreated or treated with puromycin (200 µM for 2h and 30 mins). Whole-cell lysates were analyzed by SDS–PAGE and immunoblotting. CCDC47 was used as a loading control. **(E)** Two-dimensional electrophoresis analysis (BN-PAGE/SDS–PAGE) of SYNJ2BP-interacting proteins in fibroblasts overexpressing SYNJ2BP untreated or treated with puromycin

stimulation (Fig 5B), suggesting that the observed anomalies are specifically because of MERC defects. As loss of ESYT1 does not impact SOCE in HeLa cells (Giordano et al, 2013; Woo et al, 2020), we measured total ER $Ca^{2+}$ store using the cytosolic-targeted R-GECO $Ca^{2+}$ probe upon thapsigarin treatment, an inhibitor of the sarco/ER $Ca^{2+}$ ATPase SERCA that blocks $Ca^{2+}$ pumping into the ER (Fig 5C and D) and observed no difference in our different conditions. Finally, to confirm that these defects in mitochondrial $Ca^{2+}$ uptake were not associated with a decreased levels of the main proteins involved in mitochondrial $Ca^{2+}$ flux, we analysed their levels in ESYT1-silenced HeLa cells. Acute silencing of ESYT1 did not have appreciable effects on the levels of MCU, MICU1 or MICU2 (Fig 5E and F). Together, our results in HeLa cells show that silencing of ESYT1 leads to decreased mitochondrial calcium uptake upon ER stimulation because of a decrease of MERCs.

To investigate the role of ESYT1 in mitochondrial $Ca^{2+}$ dynamics in fibroblasts, we compared control human fibroblasts, ESYT1 KO fibroblasts, and ESYT1 KO fibroblasts expressing either ESYT1-Myc or the engineered ER–mitochondria tether (Figs 6 and S3). In contrast to the above results in Hela cells, loss of ESYT1 impaired SOCE efficiency in fibroblasts, as measured with the cytosolic probe Fluoforte, after addition of calcium chloride on thapsigargin-treated cells (Fig 6A and B). We therefore investigated the influence of ESYT1 loss on cytosolic $Ca^{2+}$ concentration after ATP (Fig 6F–H) or histamine (Fig S3D–F) stimulation using the cytosolic-targeted $Ca^{2+}$ probe reporter aequorin. Both conditions showed a reduced cytosolic $Ca^{2+}$ concentration in ESYT1 KO cells after ER-$Ca^{2+}$ release. In addition, whereas ESYT1 KO does not influence the total ER $Ca^{2+}$ pool (Fig 6K and L), the decrease of ER-$Ca^{2+}$ release capacity we observed was confirmed using the ER-targeted R-CEPIA1er upon histamine stimulation (Fig 6I and J). Nevertheless, loss of ESYT1 decreased the $Ca^{2+}$ uptake capacities of mitochondria upon histamine (Fig S3A–C) or ATP stimulation (Fig 6C–E). To determine if the defect of mitochondrial $Ca^{2+}$ was fully because of the observed impairment of SOCE, or if it was partially associated with MERC defects, we performed different rescue conditions experiments. Significantly, whereas both the cytosolic and mitochondrial $Ca^{2+}$ defects were rescued by reexpression of ESYT1–Myc in ESYT1–KO fibroblsasts, expression of the artifical tether only specifically rescued the mitochondrial $Ca^{2+}$ phenotype, but not the cytosolic ones. Thus, these results suggest that similar to HeLa cells, the decrease of mitochondrial $Ca^{2+}$ uptake observed in fibroblasts is not fully because of SOCE and cytosolic $Ca^{2+}$ defects, but rather to the decrease of MERCs induced by loss of ESYT1. Finally, immunoblot analysis (Fig 6M and N) in ESYT1 KO fibroblasts showed that the levels of the major proteins involved in mitochondrial $Ca^{2+}$ pumping were not affected, nor was the assembly of the IP3R or the MCU complexes (Fig 6O). Several posttranslational modifications are known to regulate IP3R activity (Hamada & Mikoshiba, 2020) and it is possible that these could be affected by the loss of ESYT1.

Together, these results highlight the distinct and dual roles of ESYT1 in $Ca^{2+}$ regulation at the ER–PM and at MERCs.

## SYNJ2BP is required for ER to mitochondria $Ca^{2+}$ transfer

Based on the results obtained for ESYT1 and the significant increase of MERCs upon the overexpression of the OMM ESYT1 partner SYNJ2BP, we next investigated the role of SYNJ2BP in mitochondrial $Ca^{2+}$ dynamics (Fig 7). To do so, we compared control fibroblasts with SYNJ2BP KO human fibroblasts (two different clones) and fibroblasts overexpressing SYNJ2BP (either bulk cultures or a clone) (Fig 7). Similar to ESYT1 loss, the absence of SYNJ2BP strongly decreased both maximal mitochondrial $Ca^{2+}$ concentration (Fig 7A and B) and mitochondrial $Ca^{2+}$ uptake rate (Fig 7C). SYNJ2BP overexpression however significantly increased mitochondrial $Ca^{2+}$ uptake capacity upon histamine stimulation (Fig 7A–C). In contrast to ESYT1, the level of SYNJ2BP did not influence cytosolic $Ca^{2+}$ concentration (Fig 7D–F) upon histamine stimulation. Finally, SYNJ2BP overexpression did not affect levels of proteins involved in mitochondrial $Ca^{2+}$ pumping (Fig 7G and H).

To better understand the effect of SYNJ2BP on mitochondrial $Ca^{2+}$ uptake, we analyzed its role in MERC formation using an in situ proximity ligation assay (PLA), an established method to analyze MERCs (Fig 7I and J) (Tubbs & Rieusset, 2016). As seen in our TEM analysis (Fig 3A and B), overexpression of SYNJ2BP increased the number of MERCs, monitored by the increase of the number of PLA *foci* per cell compared with controls. In contrast, SYNJ2BP KO led to a reduction in the number of PLA *foci* per cell, indicating a decrease number of MERCs (Fig 7I and J). Together these results confirm that the quantity of MERCs is proportional to the level of SYNJ2BP expression (Ilacqua et al, 2022; Pourshafie et al, 2022), which therefore strongly influences mitochondrial $Ca^{2+}$ uptake capacity.

## ESYT1 regulates mitochondrial lipid homeostasis

Mitochondrial lipid composition is distinct from that in other organelles (Funai et al, 2020) and plays a critical role in the regulation of mitochondrial and cellular homeostasis (Sassano et al, 2022; Ventura et al, 2022). The most abundant mitochondrial phospholipids are phosphatidylcholine (PC), phosphatidylethanolamine (PE), cardiolipin (CL), phosphatidylinositol (PI), and phosphatidylserine (PS). CL and PE are synthetized in the IMM, requiring the import of precursor lipids, phosphatidic acid (PA) and PS, respectively, from the ER membrane at MERCs. Indeed, numerous studies have highlighted the critical contribution of MERCs in generating a platform for efficient lipid exchanges between the two organelles (Tamura et al, 2020).

As the ESYT1–SYNJ2BP complex controls MERC architecture, we investigated the role of ESYT1 in lipid transfer from ER to mitochondria. We performed shotgun mass spectrometry lipidomics, allowing broad coverage of lipids and absolute quantification (Lipotype GmbH), from purified mitochondria. We compared control human fibroblasts (control, n = 3), ESYT1 KO fibroblasts (KO, n = 4), and ESYT1 KO fibroblasts expressing either ESYT1–Myc (Rescue, n = 6) or the ER–mitochondria artificial tether (Tether, n = 6). Over 1,484 lipid entities were identified and quantified of which 149 were

---

(200 μM for 2h and 30 mins). The position of identified SYNJ2BP-containing complexes and their alignment with ESYT1 and RRBP1-containing complexes are indicated with grey lines.

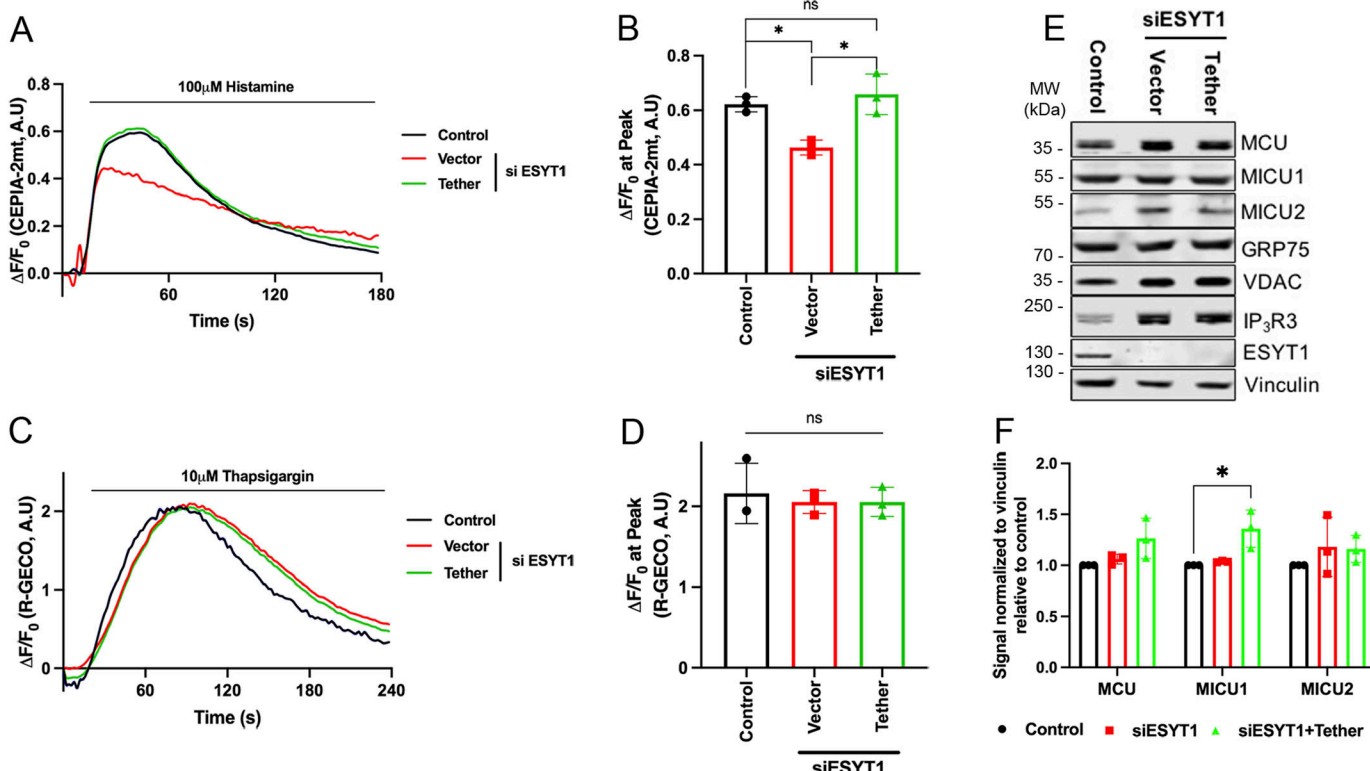

**Figure 5. ESYT1 is required for ER to mitochondria Ca$^{2+}$ transfer in Hela cells.**
**(A)** Trace of mitochondrial (Ca$^{2+}$) upon histamine stimulation (100 $\mu M$) in control HeLa cells, cells knocked-down for ESYT1, and cells knocked-down for ESYT1 that express an artificial ER–mitochondria tether. All cells express the mitochondrial Ca$^{2+}$ probe, CEPIA-2mt. **(B)** Quantification of the maximal fluorescence intensity fold-change ($\Delta F/F0$) of CEPIA-2mt induced by histamine. Results are expressed as mean ± SD; From >50 cells per condition; n = 3 independent experiments. ns: not significant; *$P < 0.05$ (Turkey's multiple comparisons test). **(C)** Trace of cytosolic (Ca$^{2+}$) upon thapsigargin treatment (10 $\mu M$) in control HeLa cells, cells knocked-down for ESYT1 and cells knocked-down for ESYT1 that express an artificial ER–mitochondria tether. All cells express the cytosolic Ca$^{2+}$ probe, R-GECO. **(D)** Quantification of the maximal fluorescence intensity fold change ($\Delta F/F0$) of R-GECO upon thapsigargin treatment. Results are expressed as mean ± SD; from >50 cells per condition; n = 3 independent experiments. ns: not significant (Turkey's multiple comparisons test). **(E)** Whole-cell lysates of control HeLa cells, cells knocked-down for ESYT1 and cells knocked-down for ESYT1 that express an artificial ER–mitochondria tether were analyzed by SDS–PAGE and immunoblotting. Vinculin was used as a loading control. **(E, F)** Quantification of three independent experiments as in panel (E). The graphs show the signal normalized to vinculin relative to control. Results are expressed as means ± S.D. Two-way ANOVA with a Dunnett correction for multiple comparisons was performed. *$P < 0.05$.

statistically different after filtering (Table S3). Multivariant data analysis using principal component analysis (Fig 8A) and hierarchical clustering with heatmap analysis (Fig S4A) showed tight clustering of the replicates and a clear separation between control, KO, and rescue conditions. ESYT1 and artificial tether overexpressing samples clustered together, suggesting that the mitochondrial lipid content is similar in these samples. Fig S4B shows the profile of the different lipid classes identified. The loss of ESYT1 resulted in a decrease proportion of the three main mitochondrial lipid categories CL, PE, and PI, which was accompanied by an increased proportion of PC (Fig 8B). Importantly, reintroduction of both ESYT1 and the artificial tether rescued this phenotype.

To investigate if overexpression of ESYT1 or the artificial tether induced ER stress, potentially changing the ER lipid composition, we performed an immunoblot analysis to compare markers of ER stress in control fibroblasts, KO ESYT1 fibroblasts, KO ESYT1 fibroblasts overexpressing ESYT1-Myc or the tether (Fig S4C). This showed no changes in the levels of several different markers of ER stress (GRP78, EIF2A, PERK) or cell death (PARP1, CAS7).

Together, these results demonstrate that ESYT1 is required for optimal lipid transfer from ER to mitochondria, likely through its tethering function as this phenotype is completely rescued by the artificial tether, suggesting that other lipid transport proteins are involved.

# Discussion

This study demonstrates that the ESYT1–SYNJ2BP tethering complex regulates essential physiological functions that occur at the mitochondrial–ER interface. ESYT1 and SYNJ2BP localize to MAM subdomains where they interact in a high molecular weight complex, favouring the formation of MERCs. The two partners are interdependent in that localization of ESYT1 at mitochondria requires SYNJ2BP expression (Fig 3C), and the absence of ESYT1 reduces the effect of SYNJ2BP overexpression on MERC induction (Fig 3E). Loss of this tethering function results in reduced mitochondrial calcium uptake capacity and impaired mitochondrial lipid homeostasis. Thus the ESYT1–SYNJ2BP complex fulfills all the essential

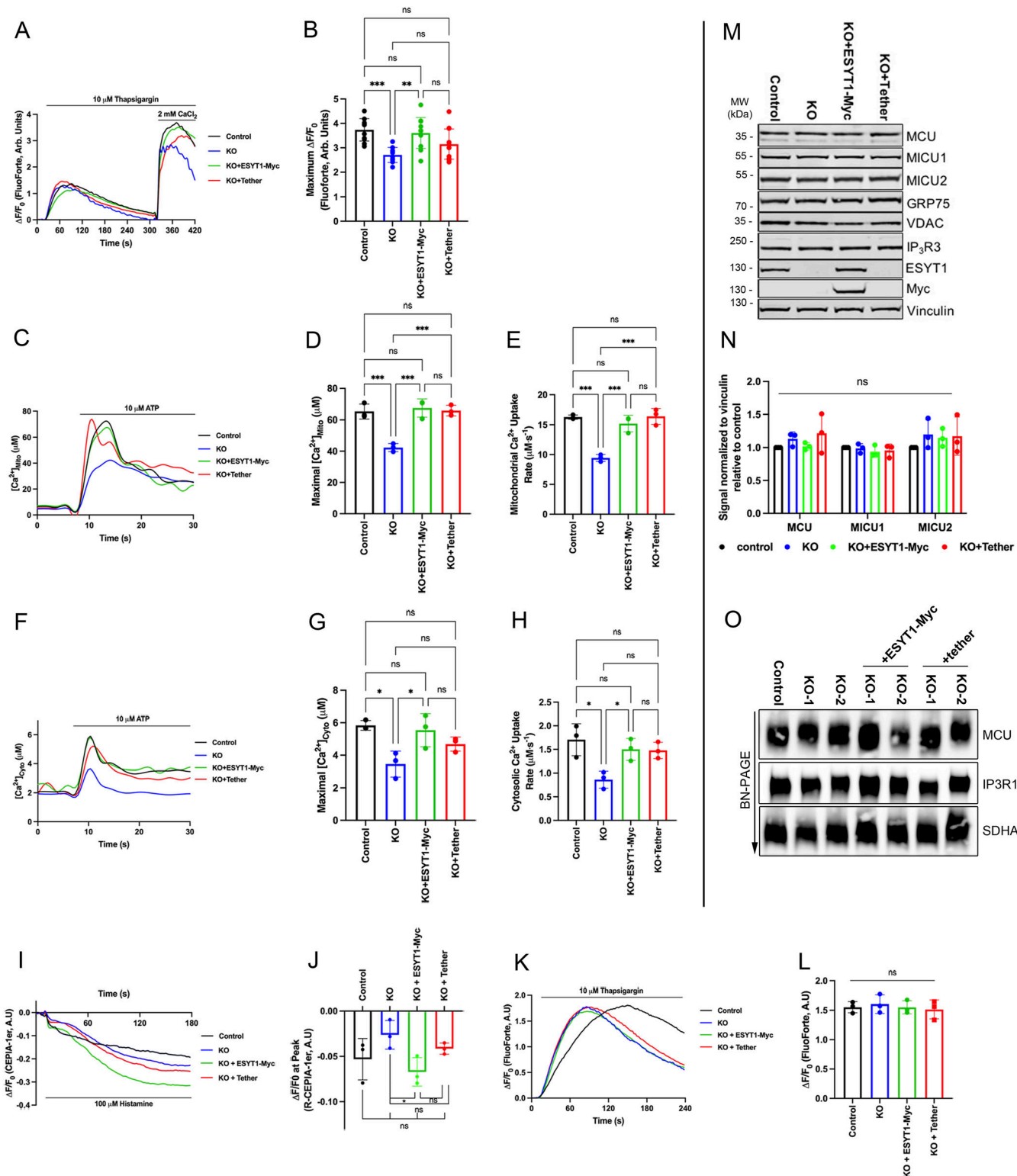

**Figure 6. ESYT1 is required for ER to mitochondria Ca²⁺ transfer in human fibroblasts.**

**(A)** Trace of cytosolic Ca²⁺ probe Fluoforte in control human fibroblasts, ESYT1 KO fibroblasts, ESYT1 KO fibroblasts expressing ESYT1–Myc, or an artificial mitochondria–ER tether, after treatment with thapsigargin (10 μM) and addition of 2 mM CaCl₂. **(B)** Quantification of maximal fold change in cytosolic Ca²⁺ levels from thapsigargin-induced ER Ca²⁺ depletion to maximal cytosolic signal in control human fibroblasts, ESYT1 KO fibroblasts, ESYT1 KO fibroblasts expressing ESYT1–Myc, or an artificial mitochondria–ER tether. Results are expressed as mean ± SD from >50 cells per condition; n = 3 independent experiments. ns: not significant; *P < 0.05; **P < 0.01; ***P < 0.001 (Turkey's multiple comparisons test). **(C)** Trace of mitochondrial–aequorin measurements of mitochondrial Ca²⁺ levels upon ATP (10 μM) stimulation in

◆◆◆◆◆ Life Science Alliance

criteria for a bona fide inter-organellar tether (Eisenberg-Bord et al, 2016; Scorrano et al, 2019). Although ESYT1 harbours calcium-binding and lipid transfer domains, both functions can be replaced at MERCs by an artificial mitochondria–ER tether.

A challenge for the study of MERCs is the multiplicity of described tethers. Although one might predict that the loss of a single protein complex would not be sufficient to disrupt MERC structure and function, that is not what we observed for ESYT1–SYNJ2BP in this study. That appears to be a general observation for the other mammalian proteins that have been proposed to tether the two organelles: PDZD8 (Hirabayashi et al, 2017), the dually OMM- and ER-localized MFN2 (de Brito & Scorrano, 2008), and the OMM protein RMDN3 that interacts with the ER protein VAPB (De Vos et al, 2012; Stoica et al, 2014). All have been shown to regulate MERC formation and loss of function in all cases can be rescued by an engineered ER-OMM linker (Gomez-Suaga et al, 2017; Hirabayashi et al, 2017; Hernández-Alvarez et al, 2019) indicating that each of these protein complexes constitutes an essential tether. Whether or how the loss of one tether affects the other tethering complexes remains unexplored, but loss of individual tethers is clearly sufficient to provoke abnormal cellular calcium dynamics and interorganellar lipid transport. These data suggest, at least in the cellular models where they have been studied, that compensatory mechanisms are not commonly up-regulated. This may not be the case in animal models. For instance, the loss of all three ESYTs does not affect mouse development, viability, fertility, brain structure, ER morphology or synaptic protein composition (Sclip et al, 2016; Tremblay & Moss, 2016), so clearly, adaptive mechanisms exist. In fact, the loss of all ESYTs induces the expression the lipid transfer proteins OSBPL5 and OSBPL8 and the SOCE-associated proteins ORAI1 and STIM1 (Tremblay & Moss, 2016). A mechanistic resolution of the interrelatedness of different tethering complexes will require further study.

The multiplicity of tether complexes also suggests the existence of different types of MERCs of variable composition, sustaining specific functions such as lipid transfer, calcium exchange or regulation of apoptosis. We demonstrated that contact sites occupied by SYNJ2BP and MFN2 are independent and are likely physically and functionally different because SYNJ2BP still promoted MERC formation in the absence of MFN2 (Fig S2B). We also show that, when overexpressed, SYNJ2BP can be part of two different complexes with ESYT1 or RRBP1 (Fig 4), that localize in different areas of the mitochondrial network (Fig 3D), suggesting that SYNJ2BP may sustain multiple functions at MERCs. Moreover, whereas the loss of either ESYT1 or SYNJ2BP reduces the number and length of MERCs, only the overexpression of SYNJ2BP enhanced MERC formation, leading to the recruitment of ESYT1 at MERCs (Fig 3C and D) and increased mitochondrial $Ca^{2+}$ uptake capacity (Fig 7). SYNJ2BP acts like a glue zipping ER to mitochondria, the quantity of MERCs being proportional to the level of SYNJ2BP expression (Fig 7). Interestingly, it has recently been reported that SYNJ2BP-dependant MERCs are involved in the physiopathology of neuronal and viral diseases (Duan et al, 2022; Pourshafie et al, 2022).

The function of ESYT1 at ER–PM contact sites has been extensively studied (Saheki, 2017). ESYT1 consists of an N-terminal hairpin-like transmembrane domain that anchors ESYT1 to the ER. The ESYT1 SMP domain binds and transports lipids in vitro (Bian et al, 2018) and the five C2 domains (A to E) bind $Ca^{2+}$ and mediate interactions with phospholipids (Corbalan-Garcia & Gomez-Fernandez, 2014). $Ca^{2+}$ binding to the C2C domain in ESYT1 enables the binding of the C2E domain to $PI(4,5)P_2$-rich membranes at the PM. It has been previously suggested that ESYT1 ER-PM tethering would be activated by and reinforce SOCE (Giordano et al, 2013; Maleth et al, 2014; Idevall-Hagren et al, 2015; Kang et al, 2019). A recent study demonstrated that ESYT1 deletion impacts SOCE in a cell-type specific manner, and that this phenotype is independent of its role in ER–PM tethering function (Woo et al, 2020). Our results in human fibroblasts confirmed that the loss of ESYT1 impairs SOCE (Fig 6). The implication of ESYT1 could then be explained by its function in the distribution and replenishment of $PIP_2$ at the ER–PM junctions (Chang et al, 2013; Maleth et al, 2014; Kang et al, 2019). Interestingly, the reintroduction of an artificial

control human fibroblasts, ESYT1 KO fibroblasts, ESYT1 KO fibroblasts expressing ESYT1-Myc, or an artificial mitochondria–ER tether. **(D)** Quantification of maximal mitochondrial $Ca^{2+}$ levels in control human fibroblasts, ESYT1 KO fibroblasts, ESYT1 KO fibroblasts expressing ESYT1–Myc or an artificial mitochondria–ER tether. Results are expressed as mean ± SD from >50 cells per condition; n = 3 independent experiments. ns: not significant; **$P$ < 0.01 (Turkey's multiple comparisons test). **(E)** Quantification of the rate of mitochondrial $Ca^{2+}$ uptake in control human fibroblasts, ESYT1 KO fibroblasts, ESYT1 KO fibroblasts expressing ESYT1–Myc or an artificial mitochondria–ER tether. Results are expressed as mean ± SD from >50 cells per condition; n = 3 independent experiments. ns: not significant; *$P$ < 0.05; **$P$ < 0.01; ***$P$ < 0.001 (Turkey's multiple comparisons test). **(F)** Representative trace of cytosolic-aequorin measurements of mitochondrial $Ca^{2+}$ levels upon ATP (10 $\mu$M) stimulation in control human fibroblasts, ESYT1 KO fibroblasts, ESYT1 KO fibroblasts expressing ESYT1–Myc or an artificial mitochondria–ER tether. Results are expressed as mean ± SD from >50 cells per condition; n = 3 independent experiments. ns: not significant; *$P$ < 0.05; **$P$ < 0.01; ***$P$ < 0.001 (Turkey's multiple comparisons test). **(G)** Quantification of maximal cytosolic $Ca^{2+}$ levels in control human fibroblasts, ESYT1 KO fibroblasts, ESYT1 KO fibroblasts expressing ESYT1–Myc or an artificial mitochondria–ER tether. Results are expressed as mean ± SD from >50 cells per condition; n = 3 independent experiments. ns: not significant; *$P$ < 0.05; **$P$ < 0.01; ***$P$ < 0.001 (Turkey's multiple comparisons test). **(H)** Quantification of the rate of cytosolic $Ca^{2+}$ uptake in control human fibroblasts, ESYT1 KO fibroblasts, ESYT1 KO fibroblasts expressing ESYT1–Myc or an artificial mitochondria–ER tether. Results are expressed as mean ± SD from >50 cells per condition; n = 3 independent experiments. ns: not significant; *$P$ < 0.05; **$P$ < 0.01; ***$P$ < 0.001 (Turkey's multiple comparisons test). **(I)** Trace of ER $Ca^{2+}$ in control human fibroblasts, ESYT1 knock-out fibroblasts, ESYT1 knock-out fibroblasts expressing either ESYT1–Myc or an artificial mitochondria–ER tether. All cell lines express the ER-targeted GECI (ER-G-CEPIA1er) fluorescent probe. ER-$Ca^{2+}$ release was stimulated with 100 $\mu$M histamine after 10 s of baseline (F/F0 ER-G-CEPIA1er). **(J)** Quantification of the fold-change in fluorescence intensity (ΔF/F0) of CEPIA-1er at the initial peak induced by histamine. Results are expressed as mean ± SD; from >50 cells per condition; n = 4 independent experiments. ns: not significant; *$P$ < 0.05 (Turkey's multiple comparisons test). **(K)** Traces of cytosolic $Ca^{2+}$ in control human fibroblasts, ESYT1–KO fibroblasts, and ESYT1–KO fibroblasts expressing either ESYT1–Myc or an artificial mitochondria–ER tether. All cell lines express the cytosolic fluorescent probe FluoForte. ER-$Ca^{2+}$ release was stimulated with 10 $\mu$M thapsigargin after 10 s of baseline (F/F0; FluoForte). **(L)** Quantification of the maximal fold change in fluorescence intensity (ΔF/F0) of FluoForte upon thapsigargin stimulation (max F/F0; FluoForte). Mean ± SD, n = 4 independent experiments. ns = not significant (Turkey's multiple comparisons test). **(M)** Whole-cell lysates of control human fibroblasts, ESYT1-KO fibroblasts and ESYT1-KO fibroblasts expressing either ESYT1-Myc or an artificial mitochondria–ER tether were analyzed by SDS–PAGE and immunoblotting. Vinculin was used as a loading control. **(M, N)** Quantification of three independent experiments as in panel (M). The graphs show the signal normalized to vinculin relative to control. Results are expressed as means ± S.D. Two-way ANOVA with a Dunnett correction for multiple comparisons was performed. ns: not significant. **(O)** Heavy membrane fractions were isolated from control human fibroblasts, ESYT1 knock-out fibroblasts, ESYT1 knock-out fibroblasts expressing ESYT1–Myc or an artificial mitochondria–ER tether, solubilized and analyzed by blue native PAGE. SDHA was used as a loading control.

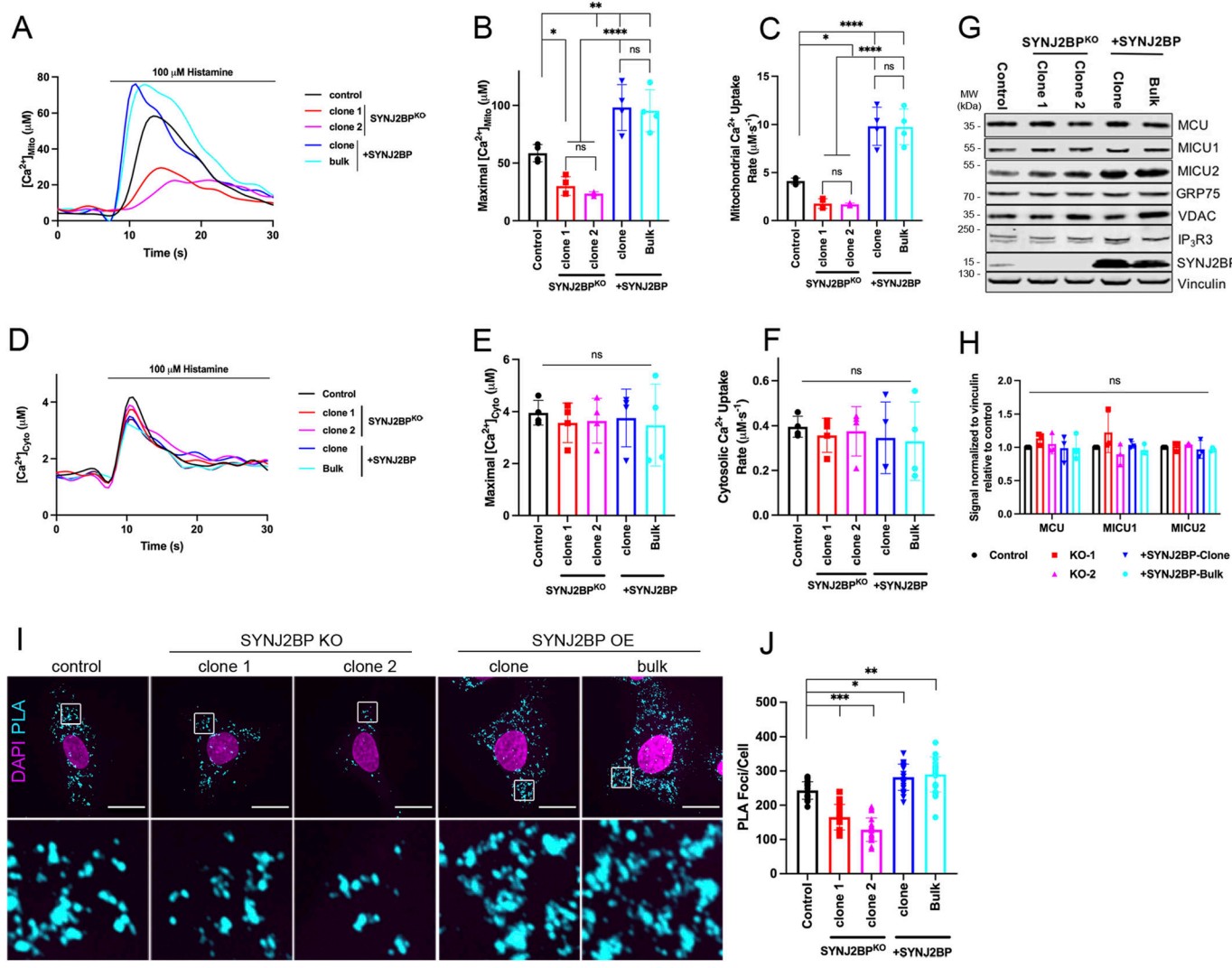

**Figure 7. SYNJ2BP is required for ER to mitochondria Ca²⁺ transfer.**
**(A)** Trace of mitochondrial–aequorin measurements of mitochondrial Ca²⁺ upon histamine stimulation (100 μM) in control human fibroblasts, SYNJ2BP knock-out fibroblasts (clone 1 and 2), and fibroblasts overexpressing SYNJ2BP (clone and bulk). **(B)** Quantification of maximal mitochondrial Ca²⁺. Results are expressed as mean ± SD. From >50 cells per condition; n = 4 independent experiments. ns: not significant; *$P < 0.05$; **$P < 0.01$; ****$P < 0.0001$ (Turkey's multiple comparisons test). **(C)** Quantification of the rate of mitochondrial Ca²⁺ uptake. Results are expressed as mean ± SD. From >50 cells per condition; n = 4 independent experiments. ns: not significant; *$P < 0.05$; ****$P < 0.0001$ (Turkey's multiple comparisons test). **(D)** Trace of cytosolic–aequorin measurements of cytosolic Ca²⁺ upon histamine stimulation (100 μM) in control human fibroblasts, SYNJ2BP knock-out fibroblasts (clone 1 and 2), and fibroblasts overexpressing SYNJ2BP (clone and bulk). **(E)** Quantification of maximal cytosolic Ca²⁺. Results are expressed as mean ± SD. From >50 cells per condition; n = 4 independent experiments. ns: not significant (Turkey's multiple comparisons test). **(F)** Quantification of the rate of cytosolic Ca²⁺ uptake. Results are expressed as mean ± SD. From >50 cells per condition; n = 4 independent experiments. ns: not significant (Turkey's multiple comparisons test). **(G)** Whole-cell lysates of control human fibroblasts, SYNJ2BP knock-out fibroblasts (clone 1 and 2), and fibroblasts overexpressing SYNJ2BP (clone and bulk) were analyzed by SDS–PAGE and immunoblotting. Vinculin was used as a loading control. **(G, H)** Quantification of three independent experiments as in panel (G). The graphs show the signal normalized to vinculin relative to control. Results are expressed as means ± S.D. Two-way ANOVA with a Dunnett correction for multiple comparisons was performed. ns: not significant. **(I)** Representative confocal images of PLA experiment in control human fibroblasts, SYNJ2BP knock-out fibroblasts (clone 1 and 2), and fibroblasts overexpressing SYNJ2BP (clone and bulk). Anti-VDAC1 and anti-IP3R1 were used as primary antibodies in the assay. Scale bars represent 20 μm. **(H, J)** Quantification of average number of PLA *foci* per cell corresponding to (H). At least 20 cells were quantified per condition per independent experiment, n = 3 independent experiments. Error bars represent mean ± SD. *$P < 0.05$, **$P < 0.01$, ***$P < 0.001$.

ER–mitochondria tether did not resolve either the cytosolic or the ER Ca²⁺ phenotype because of the loss of ESYT1, but fully rescued the mitochondrial Ca²⁺ impairment, highlighting the additional function of ESYT1 as a tether at MERCs.

Loss of ESYT1 altered mitochondrial lipid composition with significant decreases of CL, PE, and PI proportions which, in addition to being among the most abundant lipids in mitochondrial membranes (Funai et al, 2020), are essential for normal mitochondrial physiology (Belikova et al, 2006; Acin-Perez et al, 2008; Bottinger et al, 2012; Raemy & Martinou, 2014; Hsu et al, 2015; Acoba et al, 2020). The observation that the artificial tether was able to rescue this phenotype, suggests that although ESYT1 is not required for lipid transfer from ER to

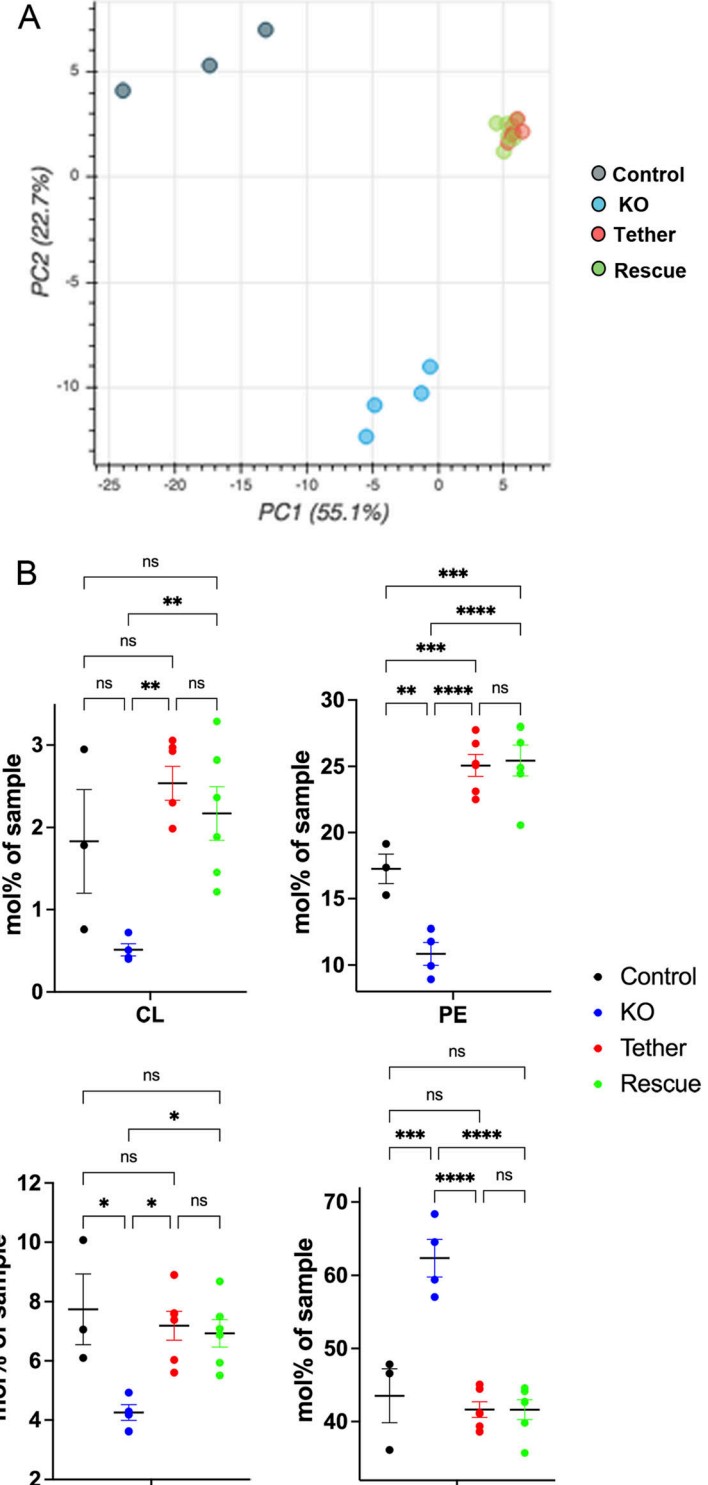

**Figure 8. ESYT1 regulates mitochondrial lipid homeostasis.**
Sucrose bilayer purified mitochondria from control human fibroblasts (control, n = 3), ESYT1 KO fibroblasts (KO, n = 4) and ESYT1 KO fibroblasts expressing either ESYT1–Myc (Rescue, n = 6) or an mitochondria–ER artificial tether (Tether, n = 6) were analyzed for absolute quantification of lipid content using shotgun mass spectrometry lipidomics. **(A)** PCA analysis of individual samples. Lipid species mol% were used as input data. **(B)** Lipid class profile of cardiolipins (CL), phosphatidylethanolamines (PE), phosphatidylinositols (PI), and phosphatidylcholines (PC). Data are presented as molar % of the total lipid amount (mol%). One-way ANOVA with multiple comparisons analysis was applied. Error bars represent mean ± SEM. ns: not significant, *P < 0.05, **P < 0.01, ***P < 0.001, ****P < 0.0001.

mitochondria, it is essential for optimal lipid transfer through its tethering property. It is possible that the mechanical tethering provided by ESYT1 might organize specialized membrane domains that serve as platforms to recruit other lipid transport proteins.

Several proteins have been proposed to participate in the lipid exchange between ER and mitochondria in mammals including RMDN3 (Yeo et al, 2021) and MFN2 (Hernández-Alvarez et al, 2019). Of particular interest, VPS13D is present at MERCs, binds the OMM

GTPase RHOT2 (Guillen-Samander et al, 2021), and has been proposed to link ER to mitochondria and support lipid transfer (Guillen-Samander et al, 2021). OSBPL5 and OSBPL8 were shown to localize to MAMs, their loss leading to mitochondrial morphology and respiration defects (Galmes et al, 2016). OSBPL5 and OSBPL8 bind to the mitochondrial intermembrane bridging/mitochondrial contact sites and cristae junction organizing system complexes, where they mediate non-vesicular transport of PS from ER to the mitochondria (Monteiro-Cardoso et al, 2022). Interestingly, we found VPS13D and VPS13A as proximity interactors of SYNJ2BP. Likewise, we found OSBPL8 as a proximity interactor of ESYT1, suggesting a potential partnership between ESYT1 as a tether and the lipid transport protein OSBPL8.

A recent study (Leterme & Michaud, 2023; Sassano et al, 2023) suggested that ESYT1 is recruited at MERCs by the ER protein PERK, independently of its kinase activity, but an OMM partner was not identified. The loss of either partner, ESYT1 or PERK, impaired ER–mitochondria lipid transfer; however, only the loss of the latter affected the quantity of MERCs and mitochondrial $Ca^{2+}$ uptake. It was concluded that ESYT1 is not involved in MERC tethering but actively transport lipids through its SMP domain. This study and ours highlight a new and previously unappreciated role of ESYT1 at MERCs and the differences between them may reflect the cellular models investigated (HeLa and shRNA-mediated knockdown vs fibroblasts and CRISPR-Cas9–mediated KO).

The molecular mechanisms that regulate SYNJ2BP–ESYT1 complex formation remain unknown. SYNJ2BP is a C-terminal tail-anchored OMM protein with a PDZ domain facing the cytosol (Hung et al, 2017). PDZ domains are small globular protein–protein interaction domains that bind the C-terminus of partner proteins. Some PDZ domains can also bind phosphatidylinositides, especially $PI(4,5)P_2$ and cholesterol (Liu & Fuentes, 2019), suggesting a synergistic binding of PDZ to phosphatidylinositide lipids and proteins (Pemberton & Balla, 2019). This raises the possibility that the binding of ESYT1 to SYNJ2BP could involve an interaction with $PI(4,5)P_2$ at the surface of the OMM, an hypothesis that will require further investigation.

# Materials and Methods

## Cell culture

Fibroblasts, HeLa cells, Flp-In T-REx 293 (Invitrogen), and Phoenix packaging (a kind gift of Garry P Nolan) cell lines were grown in 4.5 g/liter glucose DMEM (Wisent 319-027-CL) supplemented with 10% fetal bovine serum in 5% $CO_2$ incubator at 37°C. Galactose media were composed of DMEM (A14430-01; Gibco) supplemented with 10% dialysed fetal bovine serum, sodium pyruvate (Sigma-Aldrich), MEM nonessential amino acids (Gibco), GlutaMAX (Gibco), and 4.5 g/liter of galactose. Cell lines were regularly tested for mycoplasma contamination. For cytosolic translation inhibition, cells were treated with puromycin at 200 $\mu$M final concentration for 2.5 h. ON-TARGETPlus SMARTPool siRNA (Dharmacon) were used for transient knockdown of *DRP1* (L-012092-00-0005) and *MFN2* (L-012961-00-0005) and stealth siRNA (Invitrogen) for knockdown of *SYNJ2BP*

(HSS124399), *RRBP1* (HSS109381), and *ESYT1* (HSS146329). siRNAs were transiently transfected into cells using Lipofectamine RNAiMAX (Invitrogen), according to the manufacturer's specifications. Cells were analyzed after 6 d.

## Generation of KO and overexpression cell lines

KO cell lines of ESYT1 and SYNJ2BP were generated by CRISPR-Cas9–mediated gene editing in human fibroblast cells. Gene-specific target sequence 5′GTTCTTTCTCGTCGCGGACC-3′ for *ESYT1* and 5′GAAGAGATCAATCTTACCAG-3′ for *SYNJ2BP* was cloned into pSpCas9(BB)-2A-Puro (PX459) V2.0 (62988; Addgene) (Ran et al, 2013) and transfected into cells by Lipofectamine 3000 (Thermo Fisher Scientific) according to the manufacturer's instructions. The day after, transfected cells were selected by the addition of puromycin (2.5 $\mu$g/ml) for 2 d. Individual clones were screened for loss of target protein by immunoblotting and frameshift mutations were confirmed by genomic sequencing. Cells stably overexpressing ESYT1-3xFLAG, ESYT1-Myc, SYNJ2BP, and the artificial tether (Hirabayashi et al, 2017) were engineered by retroviral infection of virus produced in Phoenix cells transfected with pLXSH-Hygro plasmids as described previously (Weraarpachai et al, 2009). The artificial tether plasmid (blue fluorescent protein with OMM-targeting sequence of mAKAP1 at the N-terminus and the ER-targeting sequence of yUBC6 at the C-terminus) was a kind gift from Franck Polleux, and was engineered based on the original artificial tether from Csordas et al (2006). Flp-In T-REx 293 stable cell lines were generated as previously described (Antonicka et al, 2020).

## Bait cloning

All constructs were generated using Gateway cloning into a suitable pDEST-pcDNA5-BirA*-FLAG construct (to create either an N- or C-terminal BirA*-FLAG fusion proteins). Gateway entry clones for *ESYT1* (cat. # HOC21918; GeneCopoeia), *ESYT2* (#66831; Addgene), *PDZD8* (HsCD00400023; DNasu), and *TEX2* (HsCD00351688; DNasu) were used. For *SYNJ2BP*, an entry clone was created by PCR amplification of the ORF from human cDNA (fwd primer: 5′-GGGGACAAGTTTGTA-CAAAAAAGCAGGCTTCATGAACGGAAGAGTGGATTATTTG-3′, rev primer: 5′- GGGGACCACTTTGTACAAGAAAGCTGGGTTCAAAGTTGTTGCCGGTATCT-3′), followed by a subcloning into pDONR-221 (Invitrogen). For creation of tether_BirA* construct, the blue fluorescent protein sequence in the artificial tether was replaced with BirA*-FLAG, and the construct was cloned into pDEST-pcDNA5. For OMM_BirA* and ER_BirA*, the ER-targeting sequence (yUBC6) or the OMM-targeting sequence (mAKAP1) were removed from the tether_BirA* using mutagenesis primers 5′-CATACTCGAGATCCTTCTTTCG-3′ and 5′-CACCTACTCAGA-CAATGCGATGC-3′, respectively.

For selection of stable Flp-In T-REx 293 expressing clones, a previously described procedure was used, and representative images for all baits are shown in Fig S1 (Antonicka et al, 2020).

## Immunofluorescence

For immunofluorescence experiments, cells plated on coverslips 24 h before the experiment were fixed using 4% formaldehyde in PBS for 20 min at 37°C. Coverslips were washed three times with PBS

and cells were permeabilized in 0.1% Triton in PBS for 15 min at room temperature. After three washes with PBS, coverslips were blocked in PBS containing 5% BSA for 30 min, incubated with primary antibodies for 1 h at room temperature, washed three times with PBS, and incubated with Alexa-conjugated secondary antibodies (1:2,000) and DAPI (1:2,000) for 30 min at room temperature. Coverslips were washed three times with PBS and mounted with Fluromount-G (Thermo Fisher Scientific). Cells were imaged with Olympus IX83 microscope connected with Yokogawa CSU-X confocal scanning unit, using UPLANSAPO 100x/1.40 Oil objective (Olympus) and Andor Neo sCMOS camera. Images were processed in Fiji (Schindelin et al, 2012).

## BioID sample preparation, mass-spec data acquisition, and MS data analysis

BioID analysis, mass spectra acquisition, and MS data analysis were performed as described previously (Antonicka et al, 2020). For analysis with SAINT, only proteins with iProphet protein probability >0.95 were considered, which corresponds to an estimated protein level FDR of ~0.5%. A minimum of two detected peptide ions was required. SAINTexpress analysis was performed using version exp3.6.3 with two biological replicates per bait. SAINT analysis included 50 negative control runs used previously in a study by Antonicka et al (2020) consisting of untransfected Flp-In T-Rex 293 cells (to detect endogenously biotinylated proteins) and BirA*-FLAG-GFP cells (to detect preys that become promiscuously biotinylated). A threshold of 1% Bayesian false discovery rate was used to select high-confidence proximity interactors (Table S1). All nonhuman protein contaminants were removed from the SAINT file.

## Databases used for analysis

Mitocarta 3.0 (Rath et al, 2021) was used for annotation of detected preys as mitochondrial proteins. PANTHER17.0 database was used for Gene Ontology annotations (GO database released 22/03/2022).

## BioID data visualization

BioID data were visualized using ProHits-viz (Knight et al, 2017) analysis tool. For all analyses, average spectrum (AvgSpec) was used as the abundance measure and subtraction of the spectral counts across the controls was performed. The spectral counts for each prey were normalized to the Prey Sequence Length.

For ESYT1 specificity plot, a file combining BioID data of all SMP-domain proteins was used as input file and the specificity module was used. For ESYT1 versus ER_BirA* comparison plot, a file combining BioID data of all SMP-domain proteins, ER_BirA*, OMM_BirA*, and tether_BirA* was used as the input file and the Condition–condition module was used. For dot plot graph, a file combining BioID data of ESYT1, SYNJ2BP, ER_BirA*, OMM_BirA*, and tether_-BirA* was used. The figures were annotated and color-coded using the visualization module of ProHits-viz. Venn diagrams were created using either Venny 2.1 (https://bioinfogp.cnb.csic.es/tools/venny/index.html) or https://bioinformatics.psb.ugent.be/webtools/Venn/.

## ESYT1-FLAG immunoprecipitation

Heavy membrane fraction from human fibroblasts overexpressing ESYT1-Flag was lysed in lysis buffer (10 mM Tris pH 7.5, 150 mM NaCl, 1% DDM + protease inhibitor) for 20 min at 4°C, centrifuged for 15 min at 20,000g and supernatant was collected. This extract was precleared overnight at 4°C with rotational mixing with rinsed naked beads (Dynabeads Protein A; Invitrogen). Beads for immnunoprecipitation were incubated overnight at 4°C with rotational mixing with the Flag antibody in Na-phosphate pH 8, 0.08% tween20 buffer, washed three times with 0.1 M Na-phosphate/0.08% Tween 20 pH 8 buffer, and washed two times with 0.2 M TEA/0.08% Tween 20 pH 8. Antibody was crossed-linked to the beads using DMP (dimethyl pimelimidate dihydrochloride) in 0.2 M TEA/0.08% Tween 20 pH 8 (5.4 mg/ml) for 30 min with rotational mixing at room temperature. Reaction was stopped by adding 50 mM Tris/0.08% Tween 20 pH 7.5 and incubate for 15 min at room temperature with rotational mixing. Beads were washed three times with PBS/0.08% Tween 20 pH 8, not cross-linked antibody was removed by eluting twice with 0.1 M glycine/0.08% Tween 20 pH 2.5 and rotational mixing at room temperature for 10 min each time. Beads were finally washed three times with PBS/0.08% Tween 20 pH 8 and incubated with the precleared extract overnight at 4°C with rotational mixing. Naked beads treated the same way were used for negative control. Beads were then washed two times with lysis buffer, two times with high salt buffer (10 mM Tris pH 7.5, 450 mM NaCl, 0.1% DDM), and two times with low salt buffer (10 mM Tris pH 7.5, 150 mM NaCl, 0.1% DDM). Immunoprecipitated proteins were eluted twice with 0.1 M glycine/0.5% DDM pH 2.5 at 50°C for 15 min. Physiological pH was restored by adding 1MTris pH 7.5. Proteins were precipitated with trichloroacetic acid and sent for mass spectrometry analysis on an Orbitrap (Thermo Fisher Scientific) at the Institute de Recherches Cliniques de Montreal.

## Mouse liver fractionation

C57/BL6N male mice were obtained from Jackson Laboratories, and liver harvesting and animal handling were approved and performed in accordance with the Montreal Neurological Institute Animal Care Committee regulations. The fractionation was performed as described in the study by Aaltonen et al (2022).

## Heavy-membrane preparation and sucrose bilayer mitochondrial purification

For heavy-membrane fraction preparation, cells were rinsed twice, resuspended in ice-cold ST buffer (250 mM sucrose, 10 mM Tris–HCl pH 7.4) + Complete protease inhibitor cocktail (Roche), and homogenized with 10 passes of a prechilled, zero-clearance homogenizer (Kimble/Kontes). A postnuclear supernatant was obtained by centrifugation of the samples twice for 10 min at 600g. Heavy membranes were pelleted by centrifugation for 10 min at 10,000g and washed once in the same buffer. Protein concentration was determined by Bradford assay.

For sucrose bilayer mitochondrial purification, heavy-membrane fractions were resuspended in ST buffer, loaded on top of a sucrose bilayer (1 ml of 1 M sucrose in ST buffer on top of 1 ml of 1.7 M sucrose

in ST buffer), and centrifuged for 40 min at 70,000$g$. The band at the sucrose bilayer intersection containing pure mitochondria was harvested, diluted in ST buffer, and centrifuged for 10 min at 12,000$g$. The pellet was then washed once with ST buffer. Protein concentration was determined by Bradford assay.

## SDS–PAGE, BN-PAGE, two-dimensional electrophoresis, and Western blot

Blue-Native PAGE (BN-PAGE) was used to separate individual protein complexes. Heavy membranes were solubilized with 1% dodecyl maltoside or 8 mg/ml of digitonin for MCU and IP3R complexes. Solubilized samples (10–20 µg) were run in the first dimension on 6–15% polyacrylamide gradient gels as described in detail previously (Leary & Sasarman, 2009). For the second-dimension analysis, BN-PAGE/SDS–PAGE was carried out as detailed previously (Antonicka et al, 2003).

SDS–PAGE was used to separate denatured whole-cell extracts, heavy membranes or mouse fractionation samples. In general, whole cells were extracted with 1.5% lauryl maltoside in PBS, after which, 20 µg of protein was run on either 10%, 12%, or 15% polyacrylamide gels.

Separated proteins were transferred to a nitrocellulose membrane (PALL), and subsequently incubated with indicated primary and secondary antibodies in 5% skim-milk Tris-buffered saline solution with 0.1% Tween 20.

## TEM analysis

Cells were washed in 0.1 M Na cacodylate washing buffer (Electron Microscopy Sciences) and fixed in 2.5% glutaraldehyde (Electron Microscopy Sciences) in 0.1 M Na cacodylate buffer overnight at 4°C. Cells were then washed three times in 0.1 M Na cacodylate washing buffer for a total of 1 h, incubated in 1% osmium tetroxide (Mecalab) for 1 h at 4°C, and washed with ddH$_2$O three times for 10 min. Then, dehydration was performed in a graded series of ethanol/deionized water solutions from 30% to 90% for 8 min each, and 100% twice for 10 min each. The cells were then infiltrated with a 1:1 and 3:1 Epon 812 (Mecalab):ethanol mixture, each for 30 min, followed by 100% Epon 812 for 1 h. Cells were embedded in the culture wells with 100% Epon 812 and polymerized overnight in an oven at 60°C. Polymerized blocks were trimmed and 100-nm ultrathin sections were cut with an Ultracut E ultramicrotome (Reichert Jung) and transferred onto 200-mesh Cu grids (Electron Microscopy Sciences). Sections were post-stained for 8 min with 4% aqueous uranyl acetate (Electron Microscopy Sciences) and 5 min with Reynold's lead citrate (Thermo Fisher Scientific). Samples were imaged with a FEI Tecnai-12 transmission electron microscope (FEI Company) operating at an accelerating voltage of 120 kV equipped with an XR-80C AMT, 8 megapixel CCD camera. Based on the images, MERC characteristics (number, length, mitochondrial perimeter coverage) were measured using ImageJ software. The distance between ER and OMM was selected within 10–80 nm, manually traced, and quantified using ImageJ software.

## PLA

A PLA (Duolink PLA, Merk) was used to analyze the interaction of characterised ER and mitochondria resident proteins, which interact at MAMs, namely voltage-dependent anion channel1 (ab14734; Abcam) and IP3R1 (ab264281; Abcam) (Tubbs & Rieusset, 2016). Cells were cultured on coverslips in 24-well plates and were fixed in 5% PFA for 10 min at 37°C, quenched using 50 mM ammonium chloride and permeabilized with 0.1% Trition-X100 in PBS for 10 min. Between each step, cells were washed three times in PBS. Cells were blocked in Duolink blocking solution and incubated in a humidified chamber at 37°C for 1 h. Primary antibodies were diluted in Duolink antibody diluent and incubated at 4°C overnight. The next day, cells were washed twice with PBS for 5 min and probed with the appropriate secondary antibodies coupled to the template DNA strands at 37°C for 1 h at RT. The template DNA strand on each antibody was ligated by a DNA ligase at 37°C for 30 min at RT. Cells were washed twice with PBS for 5 min at RT and rolling loop DNA amplification was then initiated using a DNA polymerase and fluorescent nucleotides enabling detection by confocal microscopy. Cells were washed twice in PBS for 10 min and once in ddH$_2$O for 1 min before being mounted onto glass slides using mounting media containing 4′, 6-diamidino-2- phenylindole (DAPI) (ProLong Diamond; Invitrogen). At least 20 cells were analyzed from three independent experiments.

## Lipid extraction for mass spectrometry lipidomics

Mass spectrometry-based lipid analysis was performed by Lipotype GmbH as described (Sampaio et al, 2011). Lipids were extracted using a two-step chloroform/methanol procedure (Ejsing et al, 2009). Samples were spiked with internal lipid standard mixture containing the following: cardiolipin 16:1/15:0/15:0/15:0 (CL), ceramide 18:1; 2/17:0 (Cer), DAG 17:0/17:0, hexosylceramide 18:1; 2/12:0 (HexCer), lyso-phosphatidate 17:0 (LPA), lyso-phosphatidylcholine 12:0 (LPC), lyso-phosphatidylethanolamine 17:1 (LPE), lyso-phosphatidylglycerol 17:1 (LPG), lyso-phosphatidylinositol 17:1 (LPI), lyso-phosphatidylserine 17:1 (LPS), phosphatidate 17:0/17:0 (PA), phosphatidylcholine 17:0/17:0 (PC), phosphatidylethanolamine 17:0/17:0 (PE), phosphatidylglycerol 17:0/17:0 (PG), phosphatidylinositol 16:0/16:0 (PI), phosphatidylserine 17:0/17:0 (PS), cholesterol ester 20:0 (CE), sphingomyelin 18:1; 2/12:0; 0 (SM), triacylglycerol 17:0/17:0/17:0 (TAG). After extraction, the organic phase was transferred to an infusion plate and dried in a speed vacuum concentrator. As the first step, dry extract was resuspended in 7.5 mM ammonium acetate in chloroform/methanol/propanol (1:2:4, V:V:V) and in the second step, dry extract in 33% ethanol solution of methylamine in chloroform/methanol (0.003:5:1; V:V:V). All liquid handling steps were performed using Hamilton Robotics STARlet robotic platform with the Anti Droplet Control feature for organic solvent pipetting.

## Lipidomics MS data acquisition

Samples were analyzed by direct infusion on a QExactive mass spectrometer (Thermo Fisher Scientific) equipped with a TriVersa NanoMate ion source (Advion Biosciences). Samples were analyzed in both positive and negative ion modes with a resolution of Rm/z = 200 = 280,000 for MS and Rm/z = 200 = 17,500 for MSMS experiments, in a single acquisition. MSMS was triggered by an inclusion list encompassing corresponding MS mass ranges scanned in 1-D

increments (Surma et al, 2015). Both MS and MSMS data were combined to monitor CE, DAG, and TAG ions as ammonium adducts; PC, PC O-, as acetate adducts; and CL, PA, PE, PE O-, PG, PI, and PS as deprotonated anions. MS only was used to monitor LPA, LPE, LPE O-, LPI, and LPS as deprotonated anions; Cer, HexCer, SM, LPC, and LPC O- as acetate adducts.

## Lipidomics data analysis and post-processing

Data were analyzed with Lipotype's in-house developed lipid identification software based on LipidXplorer (Herzog et al, 2011; Herzog et al, 2012). Data post-processing and normalization were performed using Lipotype's in-house developed data management system. Only lipid identifications with a signal-to-noise ratio >5, and a signal intensity fivefold higher than in corresponding blank samples were considered for further data analysis.

## Lipidomics statistical analysis

Lipidomics result analysis was performed using the integrative tool LipotypeZoom from Lipotype. Lipids were selected with a cut-off of fold change ≥ ±3 and a $P$-value < 0.05 with a Benjamini & Hochberg adjustment.

## Aequorin-based mitochondrial and cytosolic calcium measurements

To measure cytosolic or mitochondrial $Ca^{2+}$ concentration, cells were cultured in white 96-well plates (Corning) and reverse-transduced with adenovirus containing either the mutated mitochondrial matrix-targeted (mtAEQmut) (Montero et al, 2000) or wild-type cytosolic aequorin (CytAEQ) (Brini et al, 1995) probes and incubated overnight at 37°C and 5% $CO_2$. Cells were washed three times in BSS + $Ca^{2+}$ (120 mM NaCl, 5.4 mM KCl, 0,8 mM $MgCl_2$, 6 mM $NaHCO_3$, 5.6 mM D-glucose, 2 mM $CaCl_2$, and 25 mM HEPES [pH 7.3]) and incubated with 5 $\mu$M coelenterazine (Sigma-Aldrich) in BSS + $Ca^{2+}$ for 90 min at 37°C and 5% $CO_2$. Post-incubation, cells were washed once in BSS + $Ca^{2+}$ and luminescence was measured by spectrophotometry (ClarioSTAR, BMG LabTek). Luminescence was measured every 2 s for 2 min. Basal luminescence was measured for 10 s followed by 100 $\mu$M histamine stimulation. At 1 min, cells were digitonized and saturated with $Ca^{2+}$ by injection of 100 $\mu$M digitonin and 10 mM $CaCl_2$ to discharge all luminous potential. Aequorin luminescence was calibrated into $Ca^{2+}$ concentration using Equation (1). For mtAEQmut: n = 1.43, $K_{TR}$ = 22,008 and $K_R$ = 22,770,000. For CytAEQ: n = 2.99, $K_{TR}$ = 120 and $K_R$ = 7,230,000. Statistical significance was determined from four independent experiments (N = 4) by repeated measures one-way ANOVA and Tukey's post hoc test for differences.

$$Ca^{2+}(M) = \frac{\left(\frac{L}{L_{Max}} \times \lambda\right)^{\frac{1}{n}} + \left(\left(\frac{L}{L_{Max}} \times \lambda\right)^{\frac{1}{n}} \times K_{TR}\right) - 1}{K_R - \left(\left(\frac{L}{L_{Max}} \times \lambda\right)^{\frac{1}{n}} \times K_R\right)} \qquad (1)$$

Equation (1). Relationship between $Ca^{2+}$ concentration and AEQ luminescence. $L$ = Light intensity, $L_{Max}$ = Sum of all light intensities,

$K_R$ = Constant for $Ca^{2+}$-bound state, $K_{TR}$ = Constant for $Ca^{2+}$-unbound state, $\lambda$ = Rate constant for AEQ consumption at $Ca^{2+}$ saturation. $n$ = Number of $Ca^{2+}$ binding sites (Bonora et al, 2013).

## Intracellular calcium analysis

Cells were seeded on a Nunc Lab-Tek chambered eight-well cover glass (Thermo Fisher Scientific). To measure mitochondrial, cytosolic, and ER calcium content, cells were transfected respectively with plasmids encoding mitochondria-targeted GECI (CEPIA2mt), cytosolic-targeted GECI (R-GECO) or cytosolic-targeted FluoForte and ER-targeted GECI (R-CEPIA1er) (Suzuki et al, 2014) using Fugene HD, following the manufacturer's instructions. 24 h after transfections, cells were washed three times in a BSS buffer (120 mM NaCl, 5.4 mM KCl, 0.8 mM $MgCl_2$, 6 mM $NaHCO_3$, 5.6 mM D-glucose, 2 mM $CaCl_2$, and 25 mM HEPES [pH 7.3]) before analysis. Fluorescence values were then collected every 2 s, and cells were stimulated with 10 $\mu$M histamine in BSS. Fluorescence was recorded for 3 min using the 40x objective of the Nikon Eclipse Ti-E microscope of the Andor Dragonfly spinning disk confocal system coupled with an Andor Ixon camera, exciting with a 488 nm or 568 nm laser for CEPIA-2mt/G-CEPIA1ER or R-GECO, respectively. Changes of fluorescence (ΔF) from each fluorescent calcium probe were normalized by basal signals before histamine stimulation (F0).

To analyse store-operated calcium entry (SOCE), cells were first seeded on a Nunc Lab-Tek chambered eight-well cover glass (Ibidi). The cells were washed three times in BSS − $Ca^{2+}$ and incubated in BSS − $Ca^{2+}$ for 1 h. The cells were incubated with Fluoforte (5 mM) in BSS − $Ca^{2+}$ for 15 min at 37°C. Post-incubation, cells were washed three times in BSS − $Ca^{2+}$. Fluorescence values were then collected every 5 s, ER calcium store depletion was induced through the inhibition of SERCA by thapsigargin (10 mM) at t = 0.5 min. Upon ER calcium store depletion, SOCE was activated by addition of exogenous CaCl2 (2 mM) at t = 5 min. Fluorescence was recorded for 7 min using the 40x objective of the Nikon Eclipse Ti-E microscope of the Andor Dragonfly spinning disk confocal system coupled with an Andor Ixon camera, exciting with a 488 nm laser.

# Data Availibility

Dataset consisting of raw files and associated peak lists and results files have been deposited in ProteomeXchange (http://www.proteomexchange.org, accession number PXD046094) and in MassIVE (https://massive.ucsd.edu, accession number MSV000093090). Additional files include the sample description, the peptide/protein evidence, and the complete SAINTexpress output for the dataset, and a "README" file that describes the dataset composition and the experimental procedures associated with the submission.

# Supplementary Information

# Acknowledgements

We thank Kathleen Daigneault and Isabella Straub for advice and excellent technical assistance. We thank Frank Polleux for the kind gift of the plasmid coding for the artificial ER–mitochondria tether (Hirabayashi et al, 2017). This research was supported in part by a grant from the CIHR (173437) to EA Shoubridge. J Prudent was supported by the Medical Research Council (MRC) (MRC grants MC_UU_00015/7 and MC_UU_00028/5). JL Morris was supported by an MRC-funded graduate student fellowship.

## Author Contributions

A Janer: conceptualization, data curation, formal analysis, investigation, methodology, and writing—original draft, review, and editing.
JL Morris: formal analysis, investigation, and writing—original draft, review, and editing.
M Krols: data curation and formal analysis.
H Antonicka: data curation, formal analysis, and writing—original draft, review, and editing.
MJ Aaltonen: data curation and formal analysis.
Z-Y Lin: data curation and formal analysis.
H Anand: data curation and formal analysis.
A-C Gingras: conceptualization, supervision, and writing—review and editing.
J Prudent: conceptualization, formal analysis, and writing—original draft, review, and editing.
EA Shoubridge: conceptualization, supervision, investigation, project administration, and writing—original draft, review, and editing.

## Conflict of Interest Statement

The authors declare that they have no conflict of interest.

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
