## [Reviewer comments · Life Science Alliance]

Life Science Alliance

ESYT1 tethers the ER to mitochondria and is required for mitochondrial lipid and calcium homeostasis

Alexandre Janer, Jordan Morris, Michiel Krols, Hana Antonicka, Mari Aaltonen, Zhen-Yuan Lin, Hanish Anand, Anne-Claude Gingras, Julien Prudent, and Eric Shoubridge

DOI: <https://doi.org/10.26508/lsa.202302335>

Corresponding author(s): *Eric Shoubridge, McGill University*

Review Timeline:

Submission Date:	2023-08-23
Editorial Decision:	2023-09-19
Revision Received:	2023-10-19
Editorial Decision:	2023-10-23
Revision Received:	2023-10-25
Accepted:	2023-10-26

Scientific Editor: *Eric Sawey, PhD*

Transaction Report:

Please note that the manuscript was reviewed at Review Commons and these reports were taken into account in the decision-making process at *Life Science Alliance*.

Review
COMMONS

Reviews

Review #1

This manuscript reports the results of a study of the potential involvement of the SMP-domain-containing protein ESYT1 in ER-mitochondria tethering, and Ca⁺ and lipid exchange between the two organelles. SMP-domain proteins have been shown to localize to membrane contact site and have lipid transport activity. ESYT proteins have thus far been found at ER-plasma-membrane (PM) contacts. Here, starting from a BioID screen for partners of various SMP-domain proteins, the study focuses on a potential new interaction between ER-resident ESYT-1 and the mitochondrial outer-membrane protein SYNJ2BP. Then using a host of different approaches, the study concludes with a model in which ESYT-1-SYNJ2BP interaction tethers ER and mitochondria to regulate ion and lipid exchange between the two organelles.

This model would be very novel and interesting, as ESYT proteins have thus far only been detected at ER-PM contacts. However, the data supporting it are not unambiguous, are subject to alternative interpretation, and are sometimes contrary to the interpretation that the authors make of them. A lot of the reasoning behind the interpretation seems to be based on the fact that the authors have a hypothesis of what the effect of impacting ER-mitochondria should be, a priori, and when they observe such effects, they take it as evidence that they have indeed impacted tethering, disregarding alternative hypotheses and the possibility that the same effects can be wrought by entirely different mechanisms. Thus, the manuscript takes a few steps to involve ESYT1 in ER-mitochondria contacts but fails to make a decisive point.

****Here are major points:****

1. Localization of ESYT-1 and SYNJ2BP. The claim of a localization at ER-mitochondria contacts relies on two type of assays. Light microscopy and subcellular fractionation. Concerning microscopy, while the staining pattern is obviously colocalizing with the ER (a control of specificity of staining using KO cells would nevertheless be desirable), the idea that ESYT1 foci "partially colocalized with mitochondria" is either trivial or unfounded. Every cellular structure is "partially colocalized with mitochondria" simply by chance at the resolution of light microscopy. If the meaning of the experiment is to show that ESYT1 'specifically' colocalizes with mitochondria, then this isn't shown by the data. There is no quantification that the level of colocalization is more than expected by chance, nor that it is higher than that of any other ER protein. Moreover, the author's model implies that ESYT1 partial colocalization with mitochondria is, at least partially, due to its interaction with SYNJ2BP. This is not tested.

The subcellular fractionation assays are grounded on the idea that Mitochondria-Associated (ER) Membranes (MAM) can be purified, and are enriched for proteins that localize at ER-mitochondria contacts. This idea originated in the early 90's and since then, myriad of papers has been using MAM purification, and whole MAM proteomes have been determined. Yet the evidence that MAM-enriched proteins represent bona fide ER-mitochondria-contact-enriched proteins (as can nowadays be determined by microscopy techniques) remain scarce. Here, anyway, ESYT1 fractionation pattern is identical to that of PDI, a marker of general ER, with no indication of specific MAM accumulation. For SYNJ2BP, it is different as it is more enriched in the MAM than the general mitochondrial marker PRDX3. However, PRDX3 is a matrix protein, making it a poor comparison point, since SYNJ2BP is an OMM protein.

Again, the model implies that ESYT1 and SYNJ2BP accumulation in the MAM should be dependent on each other. This is not tested.

2. ESYT1-SYNJ2BP interaction. The starting point of the paper is a BioID signal for SYNJ2BP when BioID is fused to ESYT1. One confirmation of the interaction comes in figure 4, using blue native gel electrophoresis and assessing comigration. Because BioID is promiscuous and comigration can be spurious, better evidence is needed to make this claim. This is exemplified by the fact that, although SYNJ2BP is found in a complex comigrating with RRB1, according to the BN gel, this slow migrating complex isn't disturbed by RRB1 knockdown, but is somewhat disturbed by ESYT1 knockdown. More than a change in abundance, a change in migration velocity when either protein is absent would be evidence that these comigrating bands represent the same complex.

ESYT1-SYNJ2BP interaction needs to be tested by coimmunoprecipitation of endogenous proteins, yeast-2-hybrid, in vitro reconstitution or any other confirmatory methods.

3. Tethering by ESYT1- SYNJ2BP. This is assessed by light and electron microscopy. Absence of ESYT1 decreases several metrics for ER-mitochondria contacts (whether absence of SYNJ2BP has the same effect isn't tested). This interesting phenomenon could be due to many things, including but not limited to the possibility

that "ESYT1 tethers ER to mitochondria". This statement and the respective subheading title are therefore clearly overreaching and should be either supported by evidence or removed. Indeed, absence of ESYT1 ER-PM tethering and lipid exchange could have knock-on effects on ER-mito contacts, therefore strong statements aren't supported. Moreover, the effect on ER-mitochondria contact metrics could be due to changes in ER-mitochondria contact indeed, but may also reflect changes in ER and/or mitochondria abundance and/or distribution, which favour or disfavour their encounter. Abundance and distribution of both organelles are not controlled for.

Finally, the authors repeat a finding that SYNJ2BP overexpression induces artificial ER-mitochondria tethering. Again, according to the model, this should be, at least in part, due to interaction with ESYT1. Whether ESYT1 is required for this tethering enhancement isn't tested.

4. Phenotypes of ESYT1/SYNJ2BP KD or KO. The study goes in details to show that downregulation of either protein yields physiological phenotypes consistent with decreased ER-mitochondria tethering. These phenotypes include calcium import into mitochondria and mitochondrial lipid composition.

Figure 5 shows that histamine-evoked ER-calcium release cause an increase in mitochondrial calcium, and this increase is reduced in absence of ESYT1, without detectable change in the abundance of the main known players of this calcium import. This is rescued by an artificial ER-mitochondria tether.

However, Figure 5D shows that the increase in calcium concentration in the cytosol upon histamine-evoked ER calcium release is equally impaired by ESYT1 deletion, contrary to expectation. Indeed, if the impairment of mitochondrial calcium import was due to improper ER-mitochondria tethering in ESYT1 mutant cells, one would expect more calcium to leak into the cytosol, not less. The remaining explanation is that ESYT1 knockout desensitizes the cells to histamine, by affecting GPCR signalling at the PM, something unexplored here. In any case, a decreased calcium discharge by the ER upon histamine treatment, explains the decreased uptake by mitochondria. The authors argue that ER calcium release is unaffected by ESYT1 KO, but crucially use thapsigargin instead of histamine to show it. Thus, the most likely interpretation of the data is that ESYT1 KO affects histamine signalling and histamine-evoked calcium release upstream of ER-mitochondria contacts.

The data with SYNJ2BP deletion are more compatible with decreased ER-mito contacts, as no decreased in cytosolic calcium is observed. This is compatible with the previously proposed role of SYNJ2BP in ER-mitochondria tethering, but the difference with ESYT1 rather argue that both proteins affect calcium signalling by different means, meaning they act in different pathways.

Finally, the study delves into mitochondrial lipids to "investigated the role of the SMP-domain containing protein ESYT1 in lipid transfer from ER to mitochondria". In reality, it is not ER-mitochondria lipid transport that is under scrutiny, but general lipid homeostasis, and changes in ER-PM lipids could have knock-on effects on mitochondrial lipids without the need to invoke disruptions in ER-mitochondria transfer activity. The changes observed are interesting but could be due to anything. Surprisingly, PCA analysis shows that the rescue of the knockout by the ESYT1 gene clusters with the rescue by the artificial tether, and not with the wildtype. This indicates that overexpressing either ESYT1 or a tether cause similar lipidomic changes. These could be due, for instance, to ER stress caused by protein overexpression, and not to a rescue.

In any case the data here do not support the strong statement "Together these results demonstrate that ESYT1 is required for lipid transfer from ER to mitochondria [...]".

This model would be very novel and interesting, as ESYT proteins have thus far only been detected at ER-PM contacts. However, the data supporting it are not unambiguous, are subject to alternative interpretation, and are sometimes contrary to the interpretation that the authors make of them. A lot of the reasoning behind the interpretation seems to be based on the fact that the authors have a hypothesis of what the effect of impacting ER-mitochondria should be, a priori, and when they observe such effects, they take it as evidence that they have indeed impacted tethering, disregarding alternative hypotheses and the possibility that the same effects can be wrought by entirely different mechanisms. Thus, the manuscript takes a few steps to involve ESYT1 in ER-mitochondria contacts but fails to make a decisive point.

Review #2

The work of Janer and al. investigates the role of E-Syt1, a well known lipid transfer protein tethering ER and PM and ER and peroxisome, at ER-mitochondria contact sites (MERCs). E-Syt1 was identified has a putative MERCs component by proximity labeling performed from four SMP domain containing proteins. They identified the mitochondrial SYNJ2BP as a binding partner of E-Syt1 only. By different biochemical and microscopy approaches, they show that 1) E-Syt1 is located at MERCs and is involved in MERCs formation, 2) SYNJ2BP is

located at MERCs and regulate the extent of MERCs in cells, 3) E-Syt1 and SYNJ2BP are located in MAM and in the same high molecular weight complex. Then, they show that both proteins impaired ER-mitochondria Ca⁺⁺ exchange and that E-Syt1 influences mitochondrial lipid homeostasis, both phenotypes being rescued by artificial tether showing that only the tethering function of E-Syt1 is required. The proximity labelling experiments suggests SYNJ2BP as the mitochondrial partners of E-Syt1, however, from the data, it is not clear whether 1) the interaction between those proteins is direct, 2) if SYNJ2BP is necessary and sufficient to localize E-Syt1 at MERC, and 3) if MERCs extension induced by SYNJ2BP is dependent on E-Syt1. Those points are important to investigate because SYNJ2BP has already been shown to induce MERCs by interacting with the ER protein RRP1. In addition, some experiments need to be better quantified.

****Major comments:****

E-syt1/SYNJ2BP in MERCs formation: the authors provide several convincing lines of evidence that both proteins are in the same complex (proximity labelling, localization in the same complex in BN-PAGE, localization in MAM) but it is not clear in which extent the direct interaction between both proteins regulates ER-mitochondria tethering.

1. Pull down experiments or BiFC strategy could be performed to show the direct interaction between both proteins;
2. SYNJ2BP OE has already been demonstrated to increase MERCs and this being dependent on the ER binding partners RRP1 (10.7554/eLife.24463). Therefore, it would be of interest to perform OE of SYNJ2BP in KO syt1 to address the question of whether Syt1 is also required to increase MERCs.
3. The authors show that Syt1 punctate size increases when SYNJ2BP is OE (Fig3C), but this can be indirectly linked to the increase of MERCs in the OE line. Thus, it could be interesting to test if the number/shape of E-syt1 punctate located close to mitochondria decreases in KO SYNJ2B. This could really show the dependence of SYNJ2BP for E-syt1 function at MERCs.

Lipid analyses: the results of MS on isolated mitochondria clearly show that mitochondrial lipid homeostasis is affected on KO-Syt1 and rescued by expression of Syt1-Myc and artificial mitochondria-ER tether. However, p.15, the authors wrote "The loss of ESYT1 resulted in a decrease of the three main mitochondrial lipid categories CL, PE and PI, which was accompanied by an increase in PC ». As the results are expressed in mol%, this interpretation can be distorted by the fact that mathematically, if the content of one lipid decreases, the content of others will increase. I would suggest to express the results in lipid quantity (nmol)/mg of mitochondria proteins instead of mol%. This will clarify the role of E-Syt1 on mitochondrial lipid homeostasis and which lipid increase and decrease. Also it could be of high interest to have the lipid composition of the whole cells to reinforce the direct involvement of E-Syt1 in mitochondrial lipid homeostasis and verify that the disruption of mitochondrial lipid homeostasis is not linked to a general perturbation of lipid metabolism as this protein acts at different MCSs.

Role of Syt1 in mitochondria: the authors show a perturbation of ER-mito Ca exchange and mitochondrial lipid homeostasis in KO-Syt1 as well as a growth defect of cells grown on galactose media. Modification of lipid mitochondrial lipid homeostasis often leads to defect in mitochondria morphology and mitochondria respiration, usually because of defects in supercomplexes assembly. To better understand the impact of Syt1 on mitochondria morphology, the author could analyze the mitochondria morphology (size, shape, cristae) on their EM images of crt, KO and OE lines. Indeed, on OE (Fig3A), the mitochondria look bigger and with a different shape compared to crt. Also, they performed a lot of BN-PAGE. Is it possible to check whether the mitochondrial respiratory chain super-complexes are affected on Syt1 KO line compared to crt?

Quantifications: some western blots need to be quantified (Fig 5K, 6J, S3E); Fig1A: Can the author provide a higher magnification of the triple labeling and perform quantification about the proportion of E-Syt1 punctate located close to mitochondria?

****Minor comments:****

- Fig1E + text: according to the legend, the BN-PAGE has been performed on Heavy membrane fraction. Why the authors speak about complexes at MAM in the text of the corresponding figure? Is it the MAM or the heavy fraction (MAM + mito + ER...)? If BN have been performed from heavy membranes, it is not a real proof that E-syt1 is in MAMs.
- On fig3C (panel crt): it seems like SYNJ2BP dots are not co-localized with mito. Is this protein targeted to another organelle beside mitochondria?
- Fig3C: can the author show each channel alone and not only the merge to better appreciate mito and ER shape in control vs OE lines (as in fig S2)
- Fig4A: can the author provide a control of protein loading (membrane staining as example) to confirm the decrease of E-Syt1 in siSYNJ2BP?
- Fig5E/F: it is not clear to me why the expression of E-Syt1 in the KO is not able to complement the KO

phenotype for cytosolic Ca⁺⁺. Can the authors comment this.

Several mitochondrial-ER tethers as well as some proteins involved in Ca and/or lipid exchanges have been identified in mammals. E-Syt1 is well known to be located at ER-PM contact sites as well as ER-peroxisomes, and the presence of E-Syt1 at MERCs and its role in Ca⁺⁺ and lipid exchange are new exciting results further showing the versatility of this protein. The results concerning E-Syt1 in Ca⁺⁺ and lipid exchange are very convincing. In addition, the proximity labeling performed from four different SMP domain containing proteins is a highly valuable source of information for future work about interaction networks of those proteins. What is less in the study is the involvement of E-Syt1 interaction with SYNJ2BP for localization and function at MERCs and vice versa. Indeed, SYNJ2BP has already been shown to promote MERCs extension and to interact with the ER protein RRP1. Thus, it will be of interest to further investigate E-Syt1/SYNJ2BP interaction at MERCs.

Review #3

Janer et al. have identified ESYT1 as a novel tether between the ER and mitochondria (MERCs) with roles in lipid and calcium homeostasis. They discovered extended synaptotagmin (ESYT1) in a BioID screen, where it interacts with SYNJ2BP and forms a high molecular weight complex. The study addressed a lack of information at the level of mammalian cell system, where a key protein complex known from yeast (ERMES) is absent, suggesting other proteins take over this critical role. These proteins then control the production of cardiolipin and PE, two lipid types essential for the functioning of mitochondria. They contain SMP motifs as a signature domain required for lipid transport.

ESYT1 had previously been found to mediate lipid transfer at the plasma membrane and at peroxisomes, but the authors found it also localizes to MERCs. In a BioID screen, they have found numerous ER proteins with known roles in MERC tethering (e.g., EMC complex, BAP31, VAPB or TMX1). They have decided to focus on the aforementioned pair, which they demonstrate is enriched on MERCs (ESYT1) and mitochondria (SYNJ2BP), respectively, forming high molecular weight complexes, as detected by BN gels. Unlike RRP1-SYNJ2BP, this complex is not dependent on ongoing protein synthesis. Upon generation of ESYT1 KO fibroblasts, they show that this SMP protein compromises MERC formation through electron microscopy. SYNJ2BP overexpression specifically increases contacts, as again shown by EM, independent of mitochondrial dynamics.

In its present form, the manuscript accurately describes the role of the ESYT1-SYNJ2BP complex for MERCs. The study contains nice lipidomics that reinforce this point and suggest a metabolic consequence. This latter observation is, however, very basic and requires some extension by assaying respirometry. The calcium phenotype is currently not fully characterized either. Interference with SOCE remains a possibility and if true, this would compromise the statement that the complex also controls calcium signaling. Both would need to be investigated better to either confirm or reject these roles, in my opinion, an important question. Overall, the manuscript contains interesting characterization of a tether that could have important consequences for calcium signaling, which would be an exciting finding.

Main points

1. Confirming the MERC localization of ESYT1 should include some more of tethering factors as demonstrated interactors (some are mentioned above) and should not be limited to lipid homeostasis.
2. The fact that in ESYT1 KO cells both mitochondrial calcium transfer and cytosolic calcium accumulation are accompanied by decreased ER-cep1a1ER signal decay upon histamine addition suggest that the main reason for ER-mitochondria calcium transfer defects are due to impaired SOCE. Calcium-free medium and histamine are used to show that ESYT1 does not affect ER calcium content. However, if it affects SOCE, then the absence of extracellular calcium would abolish such an effect; moreover, histamine does not test for leak effects. As additional information, the authors should investigate whether ER calcium content is affected by the presence of extracellular calcium in the ko scenario using thapsigargin.
3. The authors should inhibit SOCE to test whether this mechanism is affected in ESYT1 KO and could account for observed signal differences. Excluding SOCE is critical, since any change in calcium entry from the outside would potentially negate a role of ESYT1 in mitochondrial calcium uptake.
4. The authors claim that ER-Geco measurements show that no change of ER calcium was observed. However, they use thapsigargin treatment and then get a peak, when the signal should show a decrease due to leak. This suggests they did not use ER-Geco in Figure S3C. What was measured and what does it mean?
5. The findings on growth in galactose medium are intriguing but are not accompanied by respirometry to confirm mitochondria are compromised upon ESYT1 KO.

Minor points:

1. The authors mention they measure mitochondrial uptake of "exogenous" calcium by applying histamine. They

should specify that this measures transferred calcium from the ER rather than uptake of calcium from the exterior (directly at the plasma membrane).

2. Expression levels of IP3Rs are not very indicative of any change of their activity. The authors should discuss how ESYT1 could affect their PTMs.

The study is certainly of high interest due to its implications for cell metabolism and calcium signaling. It contains very strong data on MERC formation and lipidomics. However, the calcium and metabolic aspects are currently not well developed and require improvements.

Reviewer 1:

1-Localization of ESYT1 and SYNJ2BP

The claim of a localization at ER-mitochondria contacts relies on two type of assays. Light microscopy and subcellular fractionation. Concerning microscopy, while the staining pattern is obviously colocalizing with the ER (a control of specificity of staining using KO cells would nevertheless be desirable)

We performed this control and we do not see any staining for Esyt1 (we have no included this negative figure showing the absence of staining)

the idea that ESYT1 foci "partially colocalized with mitochondria" is either trivial or unfounded

Every cellular structure is "partially colocalized with mitochondria" simply by chance at the resolution of light microscopy

If the meaning of the experiment is to show that ESYT1 'specifically' colocalizes with mitochondria, then this isn't shown by the data

There is no quantification that the level of colocalization is more than expected by chance

nor that it is higher than that of any other ER protein

Moreover, the author's model implies that ESYT1 partial colocalization with mitochondria is, at least partially, due to its interaction with SYNJ2BP. This is not tested.

We do not believe that the statement that "every" cellular structure partially localizes with mitochondria is valid, and in any case we have no performed a quantitation of Esyt1 at mitochondria, to demonstrate that it is not a chance observation (Fig. 1E)

Additional comments addressing these concerns below:

To analyze and measure MERCs parameters and functions, we used a set of validated methods described in the following specialized review articles (Eisenberg-Bord, Shai et al. 2016, Scorrano, De Matteis et al. 2019).

To support and confirm the localization of ESYT1-SYNJ2BP complex at MERCs, we performed supplementary BioID analysis using ER target BirA*, OMM targeted BirA* and ER-mitochondria tether BirA* (Table S1, Figure S1 and Figure 1 A and B). These results confirmed the specificity of the interaction of the 2 partners. ESYT1 was not identified as a prey in OMM BioID and SYNJ2BP was not identified in ER BioID, on the other hand both partners were identified in the ER-mitochondria tether BioID.

To improve our description of the partial localization of ESYT1 at mitochondria, we performed a quantitative analysis using confocal microscopy on control human fibroblasts stably overexpressing SEC61B-mCherry as an ER marker which were labelled with ESYT1 and TOMM40 for mitochondria. We measured the % of ESYT1 signal colocalizing with mitochondria and the % of mitochondria positive for ESYT1 (Figure 1E).

To demonstrate than ESYT1 partial colocalization with mitochondria is, at least partially, due to its interaction with SYNJ2BP, we performed a quantitative analysis using confocal microscopy. Human control fibroblasts, KO SYNJ2BP fibroblasts and SYNJ2BP overexpressing fibroblasts were labelled with ESYT1, TOMM40 for mitochondria and CANX for ER. We measured the % of ESYT1 signal colocalizing with mitochondria in each condition (Figure 3C). These data clearly show that ESYT1 localization depends on the expression of SYNJ2BP.

Membranes (MAM) can be purified and are enriched for proteins that localize at ER-mitochondria contacts. This idea originated in the early 90's and since then, myriad of papers has been using MAM purification, and whole MAM proteomes have been determined. Yet the evidence that MAM-enriched proteins represent bona fide ER-mitochondria-contact-enriched proteins (as can nowadays be determined by microscopy techniques) remain scarce.

We employed a diverse set of techniques (microscopy, biochemical purification of mitochondrial associated membranes, and BioID, a proximity labeling tool, immunoprecipitation) to demonstrate the enrichment of ESYT1 at MERC and its interaction with SYNJ2BP.

Here, anyway, ESYT1 fractionation pattern is identical to that of PDI, a marker of general ER, with no indication of specific MAM accumulation.

Actually this is not true. To highlight the enrichment of ESYT1, we have now quantified the ESYT1 signal in each fraction. These results show a similar fractionation pattern to the bona fide MAM resident protein SIGMAR1 (Figure 1F).

For SYNJ2BP, it is different as it is more enriched in the MAM than the general mitochondrial marker PRDX3. However, PRDX3 is a matrix protein, making it a poor comparison point, since SYNJ2BP is an OMM protein.

To further confirm the specific enrichment of SYNJ2BP in the MAM fraction compared to another outer mitochondrial membrane protein, we added the signal of the well characterized OMM protein CARD19 (Rios, Zhou et al. 2022).

Again, the model implies that ESYT1 and SYNJ2BP accumulation in the MAM should be dependent on each other. This is not tested.

As describe above, we demonstrated in Figure 3C than the accumulation of ESYT1 at mitochondria is, at least partially, dependent on the quantity of SYNJ2BP.

We moreover showed a reciprocal effect in Figure 3E. A quantitative analysis using confocal microscopy demonstrated that the effect of SYNJ2BP overexpression on MERCs formation is partially dependent of the presence of ESYT1.

2-ESYT1-SYNJ2BP interaction.

The starting point of the paper is a BioID signal for SYNJ2BP when BioID (you mean BirA we think) is fused to ESYT1. One confirmation of the interaction comes in figure 4, using blue native gel electrophoresis and assessing comigration. Because BioID is promiscuous and comigration can be spurious, better evidence is needed to make this claim. This is exemplified by the fact that, although SYNJ2BP is found in a complex comigrating with RRB1, according to the BN gel, this slow migrating complex isn't disturbed by RRB1 knockdown, but is somewhat disturbed by ESYT1 knockdown. More than a change in abundance, a change in migration velocity when either protein is absent would be evidence that these comigrating bands represent the same complex.

We showed in Figure 4C that the presence of SYNJ2BP in a complex of a similar molecular weight that ESYT1 (410KDa) is totally dependent of the presence of ESYT1, suggesting an interaction of the 2 proteins.

To confirm this interaction, in figure 4A we analyzed on BN cells overexpressing SYNJ2BP together with a 3xFlag tagged version of ESYT1. As a result of the addition of the Flag tag, the complex positive for ESYT1 shifted to a higher molecular weight. The complex positive for SYNJ2BP shifted to a similar the molecular weight. This demonstrates the interaction and dependence of the 2 partners.

ESYT1-SYNJ2BP interaction needs to be tested by coimmunoprecipitation of endogenous proteins, yeast-2-hybrid, in vitro reconstitution or any other confirmatory methods.

To confirm the interaction of the 2 partners, we performed co-immunoprecipitation of the ESYT1-3xFlag protein that we showed in Figure 1H to form complexes similar to the endogenous protein. SYNJ2BP is found as the strongest prey, followed by ESYT2 and SEC22B two described interactors of ESYT1, confirming the quality of the analysis (Table S2) (Giordano, Saheki et al. 2013, Gallo, Danglot et al. 2020).

3-Tethering by ESYT1- SYNJ2BP.

This is assessed by light and electron microscopy. Absence of ESYT1 decreases several metrics for ER-mitochondria contacts (whether absence of SYNJ2BP has the same effect isn't tested).

Using PLA (proximity ligation assay) we demonstrated that the loss of SYNJ2BP leads to a decrease in MERCs (Figure 7 H and I), confirming previous studies (Ilacqua, Anastasia et al. 2022, Pourshafie, Masati et al. 2022).

This interesting phenomenon could be due to many things, including but not limited to the possibility that "ESYT1 tethers ER to mitochondria".

This statement and the respective subheading title are therefore clearly overreaching and should be either supported by evidence or removed.

Indeed, absence of ESYT1 ER-PM tethering and lipid exchange could have knock-on effects on ER-mito contacts, therefore strong statements aren't supported.

Moreover, the effect on ER-mitochondria contact metrics could be due to changes in ER-mitochondria contact indeed but may also reflect changes in ER and/or mitochondria abundance and/or distribution, which favour or disfavour their encounter. Abundance and distribution of both organelles are not controlled for.

The mitochondrial phenotypes caused by the loss of ESYT1 are all rescued by the introduction of an artificial mitochondrial-ER tether, demonstrating that they are due to loss of the tethering function of ESYT1. Our results are consistent with the generally accepted evidence required to demonstrate bona fide ER-mitochondrial tethers (enriched at contact sites, loss of membrane proximity, rescue of phenotypes by a synthetic tether, specific interaction of an ER protein with a specific mitochondrial outer membrane protein).

Finally, the authors repeat a finding that SYNJ2BP overexpression induces artificial ER-mitochondria tethering. Again, according to the model, this should be, at least in part, due to interaction with ESYT1. Whether ESYT1 is required for this tethering enhancement isn't tested.

As described above, we demonstrated in Figure 3C that the accumulation of ESYT1 at mitochondria is, at least partially, dependent on the quantity of SYNJ2BP.

We moreover showed a reciprocal effect in Figure 3F. A quantitative analysis using confocal microscopy demonstrated that the effect of SYNJ2BP overexpression on MERC formation is partially dependent of the presence of ESYT1.

4-Phenotypes of ESYT1/SYNJ2BP KD or KO.

The study goes in details to show that downregulation of either protein yields physiological phenotypes consistent with decreased ER-mitochondria tethering. These phenotypes include calcium import into mitochondria and mitochondrial lipid composition.

Figure 5 shows that histamine-evoked ER-calcium release cause an increase in mitochondrial calcium, and this increase is reduced in absence of ESYT1, without detectable change in the abundance of the main known players of this calcium import. This is rescued by an artificial ER-mitochondria tether.

However, Figure 5D shows that the increase in calcium concentration in the cytosol upon histamine-evoked ER calcium release is equally impaired by ESYT1 deletion, contrary to expectation. Indeed, if the impairment of mitochondrial calcium import was due to improper ER-mitochondria tethering in ESYT1 mutant cells, one would expect more calcium to leak into the cytosol, not less.

The remaining explanation is that ESYT1 knockout desensitizes the cells to histamine, by affecting GPCR signalling at the PM, something unexplored here.

In any case, a decreased calcium discharge by the ER upon histamine treatment, explains the decreased uptake by mitochondria.

The authors argue that ER calcium release is unaffected by ESYT1 KO, but crucially use thapsigargin instead of histamine to show it. Thus, the most likely interpretation of the data is that ESYT1 KO affects histamine signalling and histamine-evoked calcium release upstream of ER-mitochondria contacts.

Silencing ESYT1 impairs SOCE efficiency in Jurkat cells (Woo, Sun et al. 2020), but not in HeLa cells (Giordano, Saheki et al. 2013, Woo, Sun et al. 2020). Analysis of the role of ESYT1 in HeLa cells prevents confounding effects due to the loss of ESYT1 at ER-PM. In this model, knock-down of ESYT1 led to a decrease of mitochondrial Ca^{2+} uptake from the ER upon histamine stimulation, as monitored by genetically encoded Ca^{2+} indicator targeted to mitochondrial matrix (Figure 5A and B). ESYT1 silencing in HeLa cells did not impact ER Ca^{2+} store measured by the ER-targeted R-GECO Ca^{2+} probe (Figure 5C and D). The expression of the artificial mitochondria-ER tether was able to rescue mitochondrial Ca^{2+} defects observed in ESYT1 silenced cells (Figure 5B), confirming that the observed anomalies are specifically due to MERC defects.

In contrast, loss of ESYT1 impaired SOCE efficiency in fibroblasts (Figure 6 A and B). This phenotype was fully rescued by re-expression of ESYT1-Myc but not the artificial tether. We therefore investigated the influence of ESYT1 loss on cytosolic Ca^{2+} concentration following ATP (Figure 6F to H) or histamine stimulation (Figure S3 D to F), both of which showed a reduced cytosolic Ca^{2+} concentration and uptake in ESYT1 KO cells. This phenotype was fully rescued by the re-expression of ESYT1-Myc but not the artificial tether. Measurement of cytosolic Ca^{2+} after thapsigargin treatment in Ca^{2+} -free media, an inhibitor of the sarco/endoplasmic reticulum Ca^{2+} ATPase SERCA that blocks Ca^{2+} pumping into the ER, showed that ESYT1 KO does not influence the total ER Ca^{2+} pool (Figure 6K and L). However, ER- Ca^{2+} release capacity upon histamine stimulation (Figure 6I and J) is decreased in ESYT1 KO cells. This phenotype was fully rescued by the re-expression of ESYT1-Myc but not the artificial tether. Loss of ESYT1 decreased the Ca^{2+} uptake capacities of mitochondria after activation with histamine (Figure S3 A to C) or ATP (Figure 6 C to E). This phenotype was rescued by re-expression of ESYT1-Myc and also the engineered ER-mitochondria tether. Thus, despite the ER- Ca^{2+} release defect observed after ESYT1 loss, the artificial tether fully rescued the mitochondrial phenotype.

These results highlight the distinct and dual roles of ESYT1 in Ca^{2+} regulation at the ER-PM and at MERCs.

The data with SYNJ2BP deletion are more compatible with decreased ER-mito contacts, as no decreased in cytosolic calcium is observed. This is compatible with the previously proposed role of SYNJ2BP in ER-mitochondria tethering, but the difference with ESYT1 rather argue that both proteins affect calcium signaling by different means, meaning they act in different pathways.

We explain the different results concerning cytosolic calcium by the fact that ESYT1 is a dual-localized protein with dual functions on cellular calcium, implicated both in SOCE at ER-PM and in mitochondrial calcium uptake at MERCs. On the other hand, SYNJ2BP is only present at MERCs and its loss does not influence PM-ER signaling or ER- Ca^{2+} release.

Finally, the study delves into mitochondrial lipids to "investigated the role of the SMP-domain containing protein ESYT1 in lipid transfer from ER to mitochondria". In reality, it is not ER-mitochondria lipid transport that is under scrutiny, but general lipid homeostasis, and changes in ER-PM lipids could have knock-on effects on mitochondrial lipids without the need to invoke disruptions in ER-mitochondria transfer activity.

The fact that the synthetic tether, which specifically rescues MERCs, fully rescues the mitochondrial lipid phenotype argues for a direct loss of MERC tethering function when ESYT1 is absent.

The changes observed are interesting but could be due to anything. Surprisingly, PCA analysis shows that the rescue of the knockout by the ESYT1 gene clusters with the rescue by the artificial tether, and not with the wildtype. This indicates that overexpressing either ESYT1 or a tether cause similar lipidomic changes. These could be due, for instance, to ER stress caused by protein overexpression, and not to a rescue.

In order to verify if the overexpression of ESYT1 or the artificial tether induces ER stress, we performed a WB analysis to compare markers of ER stress in control fibroblasts, KO ESYT1 fibroblasts, KO ESYT1 fibroblasts overexpressing ESYT1-Myc or the tether (Figure S4C). This showed no changes in the levels of several different markers of ER stress (enumerated in the manuscript) or cell death. In addition, analysis of specific lipid classes shows that in several cases the rescue and artificial tether are not significantly different from controls (Fig. 8B, Fig. S4B). In fact the results shown in these two figures demonstrate that the reason the rescue and synthetic tether experiments do not overlap with controls is largely due to the fact that both have increased contents of PE, which is specifically synthesized in mitochondria, relative to controls.

Reviewer 2:

1) the interaction between those proteins is direct,
2) if SYNJ2BP is necessary and sufficient to localize E-Syt1 at MERC, and
3) if MERCs extension induced by SYNJ2BP is dependent on E-Syt1.
Those points are important to investigate because SYNJ2BP has already been shown to induce MERCs by interacting with the ER protein RRBP1. In addition, some experiments need to be better quantified.

Major comments:

E-syt1/SYNJ2BP in MERCs formation: the authors provide several convincing lines of evidence that both proteins are in the same complex (proximity labelling, localization in the same complex in BN-PAGE, localization in MAM) but it is not clear in which extent the direct interaction between both proteins regulates ER-mitochondria tethering.

1- Pull down experiments or BiFC strategy could be performed to show the direct interaction between both proteins.

We showed in Figure 4C that the presence of SYNJ2BP in a complex of a similar molecular weight to that ESYT1 (410KDa) is totally dependent of the presence of ESYT1, suggesting an interaction of the 2 proteins. To confirm this interaction, in figure 4A we analyzed on BN cells overexpressing SYNJ2BP together with a 3xFlag tagged version of ESYT1. As a result of the addition of the Flag tag, the complex positive for ESYT1 shifted to a higher molecular weight. Significantly, the complex positive for SYNJ2BP shifted to a similar the molecular weight, demonstrating the interaction and dependence of the 2 protein partners.

To confirm the interaction of the 2 partners, we performed co-immunoprecipitation of the ESYT1-3xFlag protein (Table S2). SYNJ2BP was found to be the strongest prey, followed by ESYT2 and SEC22B two described interactors of ESYT1, confirming the quality of the analysis (Giordano, Saheki et al. 2013, Gallo, Danglot et al. 2020).

2- SYNJ2BP OE has already been demonstrated to increase MERCs and this being dependent on the ER binding partners RRBP1 (10.7554/eLife.24463). Therefore, it would be of interest to perform OE of SYNJ2BP in KO ESYT1 to address the question of whether ESYT1 is also required to increase MERCs.

A quantitative analysis using confocal microscopy demonstrated that the effect of SYNJ2BP overexpression on MERCs formation is partially dependent of the presence of ESYT1 (Figure 3F).

3- The authors show that Eys1 punctate size increases when SYNJ2BP is OE (Fig3C), but this can be indirectly linked to the increase of MERCs in the OE line. Thus, it could be interesting to test if the number/shape of E-syt1 punctate located close to mitochondria decreases in KO SYNJ2B. This could really show the dependence of SYNJ2BP for E-syt1 function at MERCs.

To improve our description of the partial localization of ESYT1 at mitochondria, we performed a quantitative analysis using confocal microscopy on control human fibroblasts stably overexpressing SEC61B-mCherry as an ER marker which were labelled with ESYT1 and TOMM40 for mitochondria. We measured the % of ESYT1 signal colocalizing with mitochondria and the % of mitochondria colocalizing with ESYT1 (Figure 1E).

To demonstrate that ESYT1 partial colocalization with mitochondria is, at least partially, due to its interaction with SYNJ2BP, we performed a quantitative analysis using confocal microscopy. Human control fibroblasts, KO SYNJ2BP fibroblasts and SYNJ2BP overexpressing fibroblasts were labelled with ESYT1, TOMM40 for mitochondria and CANX for ER. We measured the % of ESYT1 signal colocalizing with mitochondria in each condition (Figure 3C). These data clearly show that ESYT1 localization depends on the expression of SYNJ2BP.

Lipid analyses: the results of MS on isolated mitochondria clearly show that mitochondrial lipid homeostasis is affected on KO-Syt1 and rescued by expression of Syt1-Myc and artificial mitochondria-ER tether. However, p.15, the authors wrote "The loss of ESYT1 resulted in a decrease of the three main mitochondrial lipid categories CL, PE and PI, which was accompanied by an increase in PC ». As the results are expressed in mol%, this interpretation can be distorted by the fact that mathematically, if the content of one lipid decreases, the content of others will increase. I would suggest to express the results in lipid quantity (nmol)/mg of mitochondria proteins instead of mol%. This will clarify the role of E-Syt1 on mitochondrial lipid homeostasis and which lipid increase and decrease.

It is true that as the content of some lipids decreases others will increase, but the purpose of our experiments was to examine changes in the relative contents of the different lipid classes.

Also it could be of high interest to have the lipid composition of the whole cells to reinforce the direct involvement of E-Syt1 in mitochondrial lipid homeostasis and verify that the disruption of mitochondrial lipid homeostasis is not linked to a general perturbation of lipid metabolism as this protein acts at different MCSs.

This is beyond the scope of the manuscript, and we are not certain how this would alter the major conclusion of the manuscript. In addition, the mitochondrial lipid phenotype was rescued by the synthetic tether consistent with the fact that the mitochondrial lipid phenotype is independent of the role of ESYT1 at the plasma membrane.

To better understand the impact of Eys1 of mitochondria morphology, the author could analyze the mitochondria morphology (size, shape, cristae) on their EM images of crt, KO and OE lines. Indeed, on OE (Fig3A), the mitochondria look bigger and with a different shape compared to crt.

We did not observe obvious differences in mitochondrial morphology between control, KO and OE fibroblasts so we do not think that quantitative analysis would add to the understanding of the effect of ESYT1 on mitochondrial function.

Also, they performed a lot of BN-PAGE. Is it possible to check whether the mitochondrial respiratory chain super-complexes are affected on Eys1 KO line compared to crt?

We decided to remove the data on the metabolic consequences of ESYT1 loss since it was too preliminary, focusing instead on the effect of ESYT1 loss on calcium homeostasis.

Quantifications: some western blots needs to be quantified (Fig 5K, 6J, S3E);

We did not observe obvious differences in the protein levels so we think that quantitation would not add significantly to the understanding of the differences in calcium dynamics that we report.

Fig1A: Can the author provide a higher magnification of the triple labeling and perform quantification about the proportion of E-Syt1 punctate located close to mitochondria?

We added higher magnification of the same area in all channels and arrows that point to the foci of ESYT1 colocalizing with both ER and mitochondria (Figure 1D).

To improve our description of the partial localization of ESYT1 at mitochondria, we performed a quantitative analysis using confocal microscopy on control human fibroblasts stably overexpressing SEC61B-mCherry as an ER marker which were labelled with ESYT1 and TOMM40 for mitochondria. We measured the % of ESYT1 signal colocalizing with mitochondria and the % of mitochondria colocalizing with ESYT1 (Figure 1E).

Minor comments:

- Fig1E + text: according to the legend, the BN-PAGE has been performed on Heavy membrane fraction. Why the authors speak about complexes at MAM in the text of the corresponding figure? Is-it the MAM or the heavy fraction (MAM + mito + ER...)? If BN have been performed from heavy membranes, it is not a real proof that E-syt1 is in MAMs.

Heavy membranes have been used in this experiment. The text and conclusions have been changed accordingly.

- On fig3C (panel crt): it seems like SYNJ2BP dots are not co-localized with mito. Is this protein targeted to another organelle beside mitochondria?

SYNJ2BP is not known to be targeted organelles beside mitochondria. It is possible that those few dots outside of mitochondria could be non-specific signals from the antibody we used.

- Fig4A: can the author provide a control of protein loading (membrane staining as example) to confirm the decrease of E-Syt1 in siSYNJ2BP?

As we performed this experiment only once we have removed the statement suggesting a decrease in ESYT1 protein in response to the siSYNJ2BP.

- Fig5E/F: it is not clear to me why the expression of E-Syt1 in the KO is not able to complement the KO phenotype for cytosolic Ca⁺⁺. Can the authors comment this?

We performed further analysis using ATP to trigger calcium release from the ER (figure 6 F to H). In those conditions, expression of ESYT1 in the KO is able to complement the KO phenotype for cytosolic Ca²⁺.

Reviewer 3:

Main points

1. Confirming the MERC localization of ESYT1 should include some more of tethering factors as demonstrated interactors (some are mentioned above) and should not be limited to lipid homeostasis.

As shown in Figure 1B, VAPB, PDZD8 and BCAP31 are found as preys in the ESYT1 bioID analysis. Those proteins have been described as MERC tethers, their loss leading to mitochondrial calcium defects. To support and confirm the specificity of ESYT1-SYNJ2BP complex at MERCs, we performed a supplementary BioID analysis using ER targeted BirA* and OMM targeted BirA* (Table S1, Figure S1 and Figure 1 A and B). These results confirmed the specificity of the interaction of the 2 partners. ESYT1 was not identified as a prey in OMM BioID and SYNJ2BP was not identified in ER BioID. Additional ER-mitochondria tether BirA* analyses showed that the tether-BirA* identified both ESYT1 and SYNJ2BP as preys at MERCs, confirming the localisation of this interaction. Interestingly, a large majority of the known MERCs tethers VAPB-PTPIP51, MFN2, ITPRs, BCAP31 are also found as preys in the tether-BirA* (Figure 1B), confirming the quality of these data. To confirm the interaction of the 2 partners, we performed co-immunoprecipitation of the ESYT1-3xFlag protein. SYNJ2BP is found as the strongest prey, followed by ESYT2 and SEC22B two described interactors of ESYT1, confirming the quality of the analysis (Table S2) (Giordano, Saheki et al. 2013, Gallo, Danglot et al. 2020).

2. The fact that in ESYT1 KO cells both mitochondrial calcium transfer and cytosolic calcium accumulation are accompanied by decreased ER-cepia1ER signal decay upon histamine addition suggest that the main reason for ER-mitochondria calcium transfer defects are due to impaired SOCE. Calcium-free medium and histamine are used to show that ESYT1 does not affect ER calcium content. However, if it affects SOCE, then the absence of extracellular calcium would abolish such an effect; moreover, histamine does not test for leak effects. As additional information, the authors should investigate whether ER calcium content is affected by the presence of extracellular calcium in the ko scenario using thapsigargin.

3. The authors should inhibit SOCE to test whether this mechanism is affected in ESYT1 KO and could account for observed signal differences. Excluding SOCE is critical, since any change in calcium entry from the outside would potentially negate a role of ESYT1 in mitochondrial calcium uptake.

Silencing ESYT1 impairs SOCE efficiency in Jurkat cells (Woo, Sun et al. 2020), but not in HeLa cells (Giordano, Saheki et al. 2013, Woo, Sun et al. 2020). Analysis of the role of ESYT1 in HeLa cells prevents confounding effects due to the loss of ESYT1 at ER-PM. In this model, knock-down of ESYT1 led to a decrease of mitochondrial Ca^{2+} uptake from the ER upon histamine stimulation, as monitored by genetically encoded Ca^{2+} indicator targeted to mitochondrial matrix (Figure 5A and B). ESYT1 silencing in HeLa cells did not impact ER Ca^{2+} store measured by the ER-targeted R-GECO Ca^{2+} probe (Figure 5C and D). The expression of the artificial mitochondria-ER tether was able to rescue mitochondrial Ca^{2+} defects observed in ESYT1 silenced cells (Figure 5B), confirming that the observed anomalies are specifically due to MERC defects.

In contrast loss of ESYT1 impaired SOCE efficiency in fibroblasts (Figure 6 A and B). This phenotype was fully rescued by re-expression of ESYT1-Myc but not the artificial tether. We therefore investigated the influence of ESYT1 loss on cytosolic Ca^{2+} concentration following ATP (Figure 6F to H) or histamine stimulation (Figure S3 D to F), both of which showed a reduced cytosolic Ca^{2+} concentration and uptake in ESYT1 KO cells. This phenotype was fully rescued by the re-expression of ESYT1-Myc but not the artificial tether. Measurement of cytosolic Ca^{2+} after thapsigargin treatment in Ca^{2+} -free media, an inhibitor of the sarco/endoplasmic reticulum Ca^{2+} ATPase SERCA that blocks Ca^{2+} pumping into the ER, showed that ESYT1 KO does not influence the total ER Ca^{2+} pool (Figure 6K and L). However, ER- Ca^{2+} release capacity upon histamine stimulation (Figure 6I and J) is decreased in ESYT1 KO cells. This phenotype was fully rescued by the re-expression of ESYT1-Myc but not the artificial tether. Loss of ESYT1 decreased the Ca^{2+} uptake

capacities of mitochondria after activation with histamine (Figure S3 A to C) or ATP (Figure 6 C to E). This phenotype was rescued by re-expression of ESYT1-Myc and also the engineered ER-mitochondria tether. Thus, despite the ER-Ca²⁺ release defect observed after ESYT1 loss, the artificial tether fully rescued the mitochondrial phenotype.

These results highlight the distinct and dual roles of ESYT1 in Ca²⁺ regulation at the ER-PM and at MERCs.

4. The authors claim that ER-Geco measurements show that no change of ER calcium was observed. However, they use thapsigargin treatment and then get a peak, when the signal should show a decrease due to leak. This suggests they did not use ER-Geco in Figure S3C. What was measured and what does it mean?

We used R-GECO (not ER-GECO) which measures the cytosolic calcium.

We measured total ER Ca²⁺ store using the cytosolic-targeted R-GECO Ca²⁺ probe upon thapsigargin treatment, an inhibitor of the sarco/endoplasmic reticulum Ca²⁺ ATPase SERCA that blocks Ca²⁺ pumping into the ER (Figure 5C and D) and observed no difference in our different conditions.

5. The findings on growth in galactose medium are intriguing but are not accompanied by respirometry to confirm mitochondria are compromised upon ESYT1 KO.

We decided to remove the data on the metabolic consequences of ESYT1 loss since it was too preliminary and required further investigation, focusing instead on the effect of ESYT1 loss on calcium homeostasis

Minor points:

1. The authors mention they measure mitochondrial uptake of "exogenous" calcium by applying histamine. They should specify that these measures transferred calcium from the ER rather than uptake of calcium from the exterior (directly at the plasma membrane).

The text was clarified as suggested.

2. Expression levels of IP3Rs are not very indicative of any change of their activity. The authors should discuss how ESYT1 could affect their PTMs.

A large number of post translational modifications are known to regulate IP3R activity (Hamada and Mikoshiba 2020), and it is possible that the loss of ESYT1 could interfere with these modifications, but an exploration of this issue is beyond the scope of this study. The text was clarified as suggested.

Eisenberg-Bord, M., N. Shai, M. Schuldiner and M. Bohnert (2016). "A Tether Is a Tether Is a Tether: Tethering at Membrane Contact Sites." Dev Cell **39**(4): 395-409.

Gallo, A., L. Danglot, F. Giordano, B. Hewlett, T. Binz, C. Vannier and T. Galli (2020). "Role of the Sec22b-E-Syt complex in neurite growth and ramification." J Cell Sci **133**(18).

Giordano, F., Y. Saheki, O. Idevall-Hagren, S. F. Colombo, M. Pirruccello, I. Milosevic, E. O. Gracheva, S. N. Bagriantsev, N. Borgese and P. De Camilli (2013). "PI(4,5)P(2)-dependent and Ca(2+)-regulated ER-PM interactions mediated by the extended synaptotagmins." Cell **153**(7): 1494-1509.

Hamada, K. and K. Mikoshiba (2020). "IP(3) Receptor Plasticity Underlying Diverse Functions." Annu Rev Physiol **82**: 151-176.

Ilacqua, N., I. Anastasia, D. Alosyn, R. Ghandehari-Alavijeh, E. A. Peluso, M. C. Brearley-Sholto, L. V. Pellegrini, A. Raimondi, T. Q. de Aguiar Vallim and L. Pellegrini (2022). "Expression of Synj2bp in mouse liver regulates the extent of wrapER-mitochondria contact to maintain hepatic lipid homeostasis." Biol Direct **17**(1): 37.

Pourshafie, N., E. Masati, A. Lopez, E. Bunker, A. Snyder, N. A. Edwards, A. M. Winkelsas, K. H. Fischbeck and C. Grunseich (2022). "Altered SYNJ2BP-mediated mitochondrial-ER contacts in motor neuron disease." Neurobiol Dis: 105832.

Rios, K. E., M. Zhou, N. M. Lott, C. R. Beauregard, D. P. McDaniel, T. P. Conrads and B. C. Schaefer (2022). "CARD19 Interacts with Mitochondrial Contact Site and Cristae Organizing System Constituent Proteins and Regulates Cristae Morphology." Cells **11**(7).

Scorrano, L., M. A. De Matteis, S. Emr, F. Giordano, G. Hajnoczky, B. Kornmann, L. L. Lackner, T. P. Levine, L. Pellegrini, K. Reinisch, R. Rizzuto, T. Simmen, H. Stenmark, C. Ungermann and M. Schuldiner (2019). "Coming together to define membrane contact sites." Nat Commun **10**(1): 1287.

Woo, J. S., Z. Sun, S. Srikanth and Y. Gwack (2020). "The short isoform of extended synaptotagmin-2 controls Ca(2+) dynamics in T cells via interaction with STIM1." Sci Rep **10**(1): 14433.

September 19, 2023

Re: Life Science Alliance manuscript #LSA-2023-02335

Prof. Eric A. Shoubridge
McGill University
Montreal Neurological Institute
& Dept. of Human Genetics
McGill University
Montreal, 3801 University Street H3A 2B4
Canada

Dear Dr. Shoubridge,

Thank you for submitting your revised manuscript entitled "ESYT1 tethers the endoplasmic reticulum to mitochondria and is required for mitochondrial lipid and calcium homeostasis" to Life Science Alliance. The manuscript has been seen by the original reviewers whose comments are appended below. While the reviewers continue to be overall positive about the work in terms of its suitability for Life Science Alliance, some important issues remain.

Our general policy is that papers are considered through only one revision cycle; however, given that the suggested changes are relatively minor, we are open to one additional short round of revision. Please note that I will expect to make a final decision without additional reviewer input upon re-submission.

Please address Reviewer 2's request for quantification. The lipid analysis is not necessary for revision here.

Please submit the final revision within one month, along with a letter that includes a point by point response to the remaining reviewer comments.

To upload the revised version of your manuscript, please log in to your account: <https://lsa.msubmit.net/cgi-bin/main.plex>
You will be guided to complete the submission of your revised manuscript and to fill in all necessary information.

B. MANUSCRIPT ORGANIZATION AND FORMATTING:

Sincerely,

Reviewer #2 (Comments to the Authors (Required)):

The revised version of the manuscript addressed some of my main concerns, particularly about the interdependence of ESYT1 for SYNJ2BP localization at MERCs and vice versa and direct interaction of both proteins that was demonstrated by Co-immunoprecipitation and some quantification on confocal images. But I'm still not satisfied by two answers of the authors about quantification aspects and lipid analysis.

- Quantification: for EM or western blots, authors said they have not performed quantification because they have not seen obvious differences. To my opinion, scientific conclusions should be based on facts and not on impression they had when they saw the results. In addition, they have to keep in mind that they presented in the paper only one picture of EM or WB analyses, thus, it is definitively not obvious for the reader that there is no difference based on just one image. I don't think that making quantification of EM images or WB the authors already have is a huge time consuming task and I'm surprised the authors did not take the time to do it. Thus, this give me the impression that experiments might have been performed just once or are not reproducible. Particularly, quantification of mitochondria perimeter in ESYT1 KO/OE and SYNJ2BP OE will provide an important information about the role of ESYT1 in mitochondria biogenesis. Maybe small differences, not visible by eyes, are present. If a huge defect in mitochondrial lipid homeostasis and MERCs is observed, it could affect mitochondria morphology as already described in the literature. For WB, only one experiment is presented in Fig. 4E, 5M, 6G and in 1F, the quantification seems to have been done on one experiment as no variation is indicated. This impacts the scientific soundness of the work.

- Lipid analyses: Performing the lipidomics on the whole cell is not beyond the scope of the paper because it will help the authors to interpret properly the data in terms of perturbation of mitochondrial lipidome. If the same phenotype is observed for the whole lipidome than for the mitochondrial one, the conclusion is that ESYT1 altered general lipid homeostasis, but not specifically the mitochondrial one. If the phenotype observed in the whole lipidome is different from the mitochondrial one, it is a convincing argument to say that ESYT1 is indeed involved in mitochondrial lipid homeostasis. In addition, as I already said, looking just at the mol% can lead to misinterpretation of the results. The artificial tethering lines and complementation lines gave a similar results that is different from the WT or KO. This can be linked to different levels of contamination of the purified mitochondria by ER, as MERCs are impacted in KO, OE or artificial tethering lines, that have not been checked by WB. Thus, it is to my opinion hard to interpret data of lipidomics on isolated organelles without knowing what happened at the whole cell level and if a same level of purity is achieved between the different lines. That said, the conclusion of the authors is supported by a recent paper showing that indeed, SYT1 impacts mitochondrial lipidome without affecting the composition of the total lipid cell level.

Reviewer #3 (Comments to the Authors (Required)):

Janer et al. have revised their previous submission to Review Commons. The manuscript shows the identification of ESYT1 as a novel tether between the ER and mitochondria (MERCs) with roles in lipid and calcium homeostasis from a BioID screen. ESYT1 interacts with numerous ER proteins with known roles in MERC tethering (e.g., EMC complex, BAP31, VAPB or TMX1). Moreover, they detected specific interaction of ESYT1 with mitochondrial SYNJ2BP. This latter protein also interacts with ER-localized RRBP1, the three proteins form independent binary complexes. ESYT1 was found to localize to the ER, partially co-localizing with mitochondria. ESYT1 Ko cells showed a loss of MERCs on electron micrographs. Its over-expression with SYNJ2BP was able to increase MERCs (via EM and PLA) and SYNJ2BP can recruit ESYT1 to MERCs. These effects were found independent of Drp1 or Mfn2. ESYT1 knockdown reduced ER-mitochondria calcium transfer and differentially affects ER calcium dynamics at the plasma membrane and mitochondria. Its knockdown also compromises mitochondrial content of CL, PE and PI, while increasing PC. Remarkably, no ER stress resulted from this change of MERCs.

Much improved upon the previous version, this manuscript is beautifully written and addresses multiple aspects of MERCs, all affected to the same extent. The authors have taken a considerable effort to integrate what is known about MERCs from multiple angles. The story is therefore compact, compelling and convincing. This should make sure that this manuscript will find a wide audience and I therefore have no major concerns.

Minor points:

1. On page 13, Huang or Hung are misspelled.
2. On page 15, thapsigargin is misspelled. On page 16, fibroblasts is misspelled.

Response to the reviewers concerning the manuscript Janer et al #LSA-2023-02335

Reviewer #2:

The revised version of the manuscript addressed some of my main concerns, particularly about the interdependence of ESYT1 for SYNJ2BP localization at MERCs and vice versa and direct interaction of both proteins that was demonstrated by Co-immunoprecipitation and some quantification on confocal images. But I'm still not satisfied by two answers of the authors about quantification aspects and lipid analysis.

- Quantification: for EM or western blots, authors said they have not performed quantification because they have not seen obvious differences. To my opinion, scientific conclusions should be based on facts and not on impression they had when they saw the results. In addition, they have to keep in mind that they presented in the paper only one picture of EM or WB analyses, thus, it is definitively not obvious for the reader that there is no difference based on just one image. I don't think that making quantification of EM images or WB the authors already have is a huge time consuming task and I'm surprised the authors did not take the time to do it. Thus, this give me the impression that experiments might have been performed just once or are not reproducible. Particularly, quantification of mitochondria perimeter in ESYT1 KO/OE and SYNJ2BP OE will provide an important information about the role of ESYT1 in mitochondria biogenesis. Maybe small differences, not visible by eyes, are present. If a huge defect in mitochondrial lipid homeostasis and MERCs is observed, it could affect mitochondria morphology as already described in the literature.

As proposed by reviewer #2, we extracted data concerning the mean mitochondrial perimeter from our TEM studies. These results are now presented in figure 2C and 3B and show:

1. Figure 2C: ESYT1 KO cells have a larger mitochondrial perimeter than control cells and this phenotype is fully rescued by the expression of ESYT1-Myc. This may result from the decreased number of MERCs, the presence of which demarcates mitochondrial fission sites (Giacomello, Pyakurel et al. 2020).
2. Figure 3B: control+SYNJ2BP cells have a smaller mitochondrial perimeter than control and control+ESYT1-Flag cells, likely the result of the large increase in MERCs.

A description of these results along with their interpretation have been added in the main text, together with updated legends.

For WB, only one experiment is presented in Fig. 4E, 5M, 6G and in 1F, the quantification seems to have been done on one experiment as no variation is indicated. This impacts the scientific soundness of the work.

As proposed by reviewer #2, we quantified the levels of the proteins involved in mitochondrial Ca^{2+} pumping, namely MCU, MICU1 and MICU2. These results are now presented in figure 5F, 6N and 7H and show that neither the loss of ESYT1 nor SYNJ2BP impacts the level of these proteins. A description of these results along with their interpretation have been added in the main text, together with updated legends.

- Lipid analyses: Performing the lipidomics on the whole cell is not beyond the scope of the paper because it will help the authors to interpret properly the data in terms of perturbation of mitochondrial lipidome. If the same phenotype is observed for the whole lipidome than for the mitochondrial one, the conclusion is that ESYT1 altered general lipid homeostasis, but not specifically the mitochondrial one. If the phenotype observed in the whole lipidome is different from the mitochondrial one, it is a convincing argument to say that ESYT1 is indeed involved in mitochondrial lipid homeostasis. In addition, as I already said, looking just at the mol% can lead to misinterpretation of the results. The artificial tethering lines and complementation lines gave a similar results that is different from the WT or KO. This can be linked to different levels of contamination of the purified mitochondria by ER, as MERCs are impacted in KO, OE or artificial tethering lines, that have not been checked by WB. Thus, it is to my opinion hard to interpret data of lipidomics on isolated organelles without knowing what happened at the whole cell level and if a same level of purity is achieved between the different lines. That said, the conclusion of the authors is supported by a recent paper showing that indeed, SYT1 impacts mitochondrial lipidome without affecting the composition of the total lipid cell level.

As proposed by the editor, the lipid analysis is not necessary for revision here.

Reviewer #3:

Janer et al. have revised their previous submission to Review Commons. The manuscript shows the identification of ESYT1 as a novel tether between the ER and mitochondria (MERCs) with roles in lipid and calcium homeostasis from a BioID screen.

ESYT1 interacts with numerous ER proteins with known roles in MERC tethering (e.g., EMC complex, BAP31, VAPB or TMX1). Moreover, they detected specific interaction of ESYT1 with mitochondrial SYNJ2BP. This latter protein also interacts with ER-localized RRB1, the three proteins form independent binary complexes. ESYT1 was found to localize to the ER, partially co-localizing with mitochondria. ESYT1 Ko cells showed a loss of MERCs on electron micrographs. Its over-expression with SYNJ2BP was able to increase MERCs (via EM and PLA) and SYNJ2BP can recruit ESYT1 to MERCs. These effects were found independent of Drp1 or Mfn2. ESYT1 knockdown reduced ER-mitochondria calcium transfer and differentially affects ER calcium dynamics at the plasma membrane and mitochondria. Its knockdown also compromises mitochondrial content of CL, PE and PI, while increasing PC. Remarkably, no ER stress resulted from this change of MERCs.

Much improved upon the previous version, this manuscript is beautifully written and addresses multiple aspects of MERCs, all affected to the same extent. The authors have taken a considerable effort to integrate what is known about MERCs from multiple angles. The story is therefore compact, compelling and convincing. This should make sure that this manuscript will find a wide audience and I therefore have no major concerns.

Minor points:

1. On page 13, Huang or Hung are misspelled.
2. On page 15, thapsigargin is misspelled. On page 16, fibroblasts is misspelled.

Thanks to reviewer 3 attention, we fixed these spelling mistakes in the main text.

Giacomello, M., A. Pyakurel, C. Glytsou and L. Scorrano (2020). "The cell biology of mitochondrial membrane dynamics." Nat Rev Mol Cell Biol **21**(4): 204-224.

Moreover, according to the manuscript preparation recommendation from Life Science Alliance, we shorten the title and the abstract.

October 23, 2023

RE: Life Science Alliance Manuscript #LSA-2023-02335R

Prof. Eric A. Shoubridge
McGill University
Montreal Neurological Institute
& Dept. of Human Genetics
Montreal, 3801 University Street H3A 2B4
Canada

Dear Dr. Shoubridge,

Thank you for submitting your revised manuscript entitled "ESYT1 tethers the ER to mitochondria and is required for mitochondrial lipid and calcium homeostasis". We would be happy to publish your paper in Life Science Alliance pending final revisions necessary to meet our formatting guidelines.

- please remove the separate file with supplementary table legends, as they are already provided in the manuscript file
- please add a Category for your manuscript in our system
- please add the Twitter handle of your host institute/organization as well as your own or/and one of the authors in our system
- please use the [10 author names et al.] format in your references (i.e., limit the author names to the first 10)
- please add callouts for Figure 2B, C to your main manuscript text
- uploaded datasets should be made publicly accessible at this point. please update the Data Availability statement to remove the reviewer access information

Figure Checks:

- please add sizes next to all blots
- what are the horizontal lines indicating in Figure 4A? If these are necessary, then please indicate what these are in the figure legend
- in Figure S2A, please indicate with a box on the image where the zoomed-in version is focusing on. The magnified versions also need their own scale bars.

A. FINAL FILES:

B. MANUSCRIPT ORGANIZATION AND FORMATTING:

Sincerely,

October 26, 2023

RE: Life Science Alliance Manuscript #LSA-2023-02335RR

Prof. Eric A. Shoubridge
McGill University
Montreal Neurological Institute
& Dept. of Human Genetics
Montreal, 3801 University Street H3A 2B4
Canada

Dear Dr. Shoubridge,

Thank you for submitting your Research Article entitled "ESYT1 tethers the ER to mitochondria and is required for mitochondrial lipid and calcium homeostasis". It is a pleasure to let you know that your manuscript is now accepted for publication in Life Science Alliance. Congratulations on this interesting work.

DISTRIBUTION OF MATERIALS:

Again, congratulations on a very nice paper. I hope you found the review process to be constructive and are pleased with how the manuscript was handled editorially. We look forward to future exciting submissions from your lab.

Sincerely,
